# Giant enhancement of THz-frequency optical nonlinearity by phonon polariton in ionic crystals

Yao Lu [1,3], Qi Zhang[1,3], Qiang Wu [1✉], Zhigang Chen[1✉], Xueming Liu[2] & Jingjun Xu [1✉]

The field of nonlinear optics has grown substantially in past decades, leading to tremendous progress in fundamental research and revolutionized applications. Traditionally, the optical nonlinearity for a light wave at frequencies beyond near-infrared is observed with very high peak intensity, as in most materials only the electronic nonlinearity dominates while ionic contribution is negligible. However, it was shown that the ionic contribution to nonlinearity can be much larger than the electronic one in microwave experiments. In the terahertz (THz) regime, phonon polariton may assist to substantially trigger the ionic nonlinearity of the crystals, so as to enhance even more the nonlinear optical susceptibility. Here, we experimentally demonstrate a giant second-order optical nonlinearity at THz frequency, orders of magnitude higher than that in the visible and microwave regimes. Different from previous work, the phonon-light coupling is achieved under a phase-matching setting, and the dynamic process of nonlinear THz generation is directly observed in a thin-film waveguide using a time-resolved imaging technique. Furthermore, a nonlinear modification to the Huang equations is proposed to explain the observed nonlinearity enhancement. This work brings about an effective approach to achieve high nonlinearity in ionic crystals, promising for applications in THz nonlinear technologies.

[1] MOE Key Laboratory of Weak-Light Nonlinear Photonics, TEDA Applied Physics Institute and School of Physics, Nankai University, Tianjin 300457, China. [2] School of Automation, Nanjing University of Information Science & Technology, Nanjing 210044, China. [3] These authors contributed equally: Yao Lu and Qi Zhang. ✉email: wuqiang@nankai.edu.cn; zgchen@nankai.edu.cn; jjxu@nankai.edu.cn

Nonlinear optics has led to numerous innovations in optics and photonics[1,2], significantly advanced the technology developments in laser, spectroscopy, precision quantum metrology, and material analysis[3–9]. Recently, the field of terahertz (THz) nonlinear optics has emerged and provided powerful tools to manipulate and control solid-state materials, especially in complex condensed matter systems with strongly correlated electrons[10–12]. In 2018, it was demonstrated that graphene possesses extraordinarily high nonlinear susceptibility at THz frequency due to the presence of the hot Dirac fermions[13], where nonlinear frequency conversion led to observation of the odd-order (third, fifth and seventh) terahertz harmonics. Difference-frequency generation (DFG) of THz waves has been reported by use of near-infrared input signal, which was generated in two different kinds of phase-matching conditions by a nanosecond Nd:YAG laser[14]. Although there have been significant efforts in enhancing optical nonlinearity at THz frequencies, such as through the THz Kerr effects and second-order nonlinear effects, using GaAs, lithium niobate (LN) or other materials[12,15–17], nonlinear THz techniques and associated applications are far from mature. In fact, THz nonlinear optics is still at one of the very active and dynamically changing research frontiers.

During the past decades, many efforts have been initiated to enhance the optical nonlinearity. This typically involves increasing the pump intensity or confining the power into small regions, including the use of high-power lasers, ultrashort laser pulses, micro metal antennas, or high-quality microcavities[18–25]. However, the nonlinear processes in the THz frequency range are quite difficult to be initiated and implemented, because the peak electric field of the THz pulses generated with current technologies is still relatively low compared to that of optical laser pulses[26]. Meanwhile, the method of increasing pump intensity always brings the risk of destruction of the nonlinear materials. Therefore, improving the efficiency of nonlinear generation by enhancing the nonlinear susceptibility of the optical materials, rather than by merely increasing the laser peak intensity, becomes much desired for THz waves and other high-frequency light waves.

In recent years, it was largely realized that high mobility of the electrons could enhance nonlinear susceptibilities, and indeed strong nonlinear optical phenomena were observed in semiconductors[27] as well as in different two-dimensional (2D) materials including graphene[9,13,28,29], $MoS_2$[30], and $WSe_2$[31,32]. Other mechanisms such as dipole coupling could also lead to such enhancement[33,34]. These studies mostly focused on high frequency electromagnetic waves beyond near-infrared where only the electronic nonlinearity contributes, because ions are too heavy to respond to such fast oscillations. The experimental study of strong ionic nonlinearity was actually dated back to 1971, when the corresponding nonlinear response was achieved in the microwave spectral range[35]. It was also demonstrated that the contribution of ionic nonlinearity is significantly superior to that of the electronic one. Therefore, a natural question arises: can the phonon polariton, a coupling state of optical phonon and THz wave[36–38], be employed to trigger the ionic nonlinearity in the nonlinear crystals so as to enhance the optical nonlinearity at THz frequency?

In this article, we demonstrate experimentally a giant second-order nonlinear susceptibility at THz frequency, representing not only five orders of magnitude increase as compared to that for visible light but also three orders of magnitude larger than that for microwave. Different from the previous work on the THz generation through DFG process, both the input and output signals in our system are THz waves. Moreover, the dynamic process of nonlinear generation is directly observed in a thin-film waveguide

using a time-resolved imaging technique. We emphasize that the nonlinearity enhancement achieved here is mainly attributed to the strong phonon-light coupling rather than merely from the ionic nonlinearities, and the giant second-order nonlinearity for DFG is realized under a phase-matching configuration by judicious design of the waveguide dispersion. Furthermore, we propose a nonlinear modification to the well-known Huang equations[36,37] to explain the observed enhancement phenomenon, and show that the delocalized phonon polariton can significantly enhance the nonlinear susceptibility at THz frequency.

## Results

**Direct observation of the dynamic process of THz difference-frequency generation.** In previous experiments, THz waves have been generated by using a femtosecond laser to pump a thin LN wafer[39–42]. However, since the laser pulses travel faster than the THz waves in LN, velocity mismatch emerges in the conventional setup[41], which hampers the efficient generation of quasi-monochromatic THz waves. In our experiment, the design proposed by Lin et al is adopted and improved to generate the THz waves with two different frequencies simultaneously[43]. As shown in Fig. 1, a grating is placed in the beam path, whose first-order diffraction generates a tilted-wavefront at the entrance facet of the sample, shown by the red ellipsoid in Fig. 1a. Velocity matching is realized by carefully selecting the tilt angle $\alpha$ such that the projection of the laser velocity on the LN wafer equals to that of the THz waves. Figure 1a illustrates how the tilted-wavefront of the pump pulses matches that of the generated THz waves, and Fig. 1b shows the experimental setup (more details in Methods). The sample we used is a 50 μm-thick thin-film waveguide for THz waves[44].

The theoretical dispersion relation for the effective-phase refractive index $n_{eff}$ of the transverse electric (TE) LN waveguide modes is shown by the cyan curves in Fig. 2a[44]. Consider that a certain wavefront tilt angle $\alpha$ of the pump pulses is chosen, which determines the speed of $c/\tan\alpha$ with which the overlapping region between the pump and LN waveguide is swept over the waveguide. Then, waveguide modes are excited and travel at a phase velocity of $c/n_{eff}$, matching the sweeping velocity. Therefore, the matching condition for the generation of multi-cycle THz waves can be given by $n_{eff} = \tan\alpha$[43]. Here, the angle $\alpha$ is set

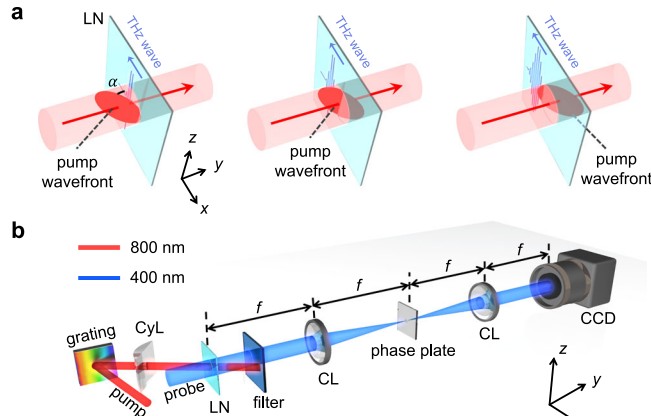

**Fig. 1 Experimental setup and illustration of velocity matching between pump pulse and generated THz wave. a** Velocity matching of the tilted-wavefront pump laser and the generated THz wave. The red ellipsoids indicate the wavefront of the pump pulse. The red arrows indicate the propagation direction of the pump pulse. **b** Experimental setup and detection of THz waves based on a pump-probe technique. CyL: cylindrical lens; LN: lithium niobate sample; CL: convex lens.

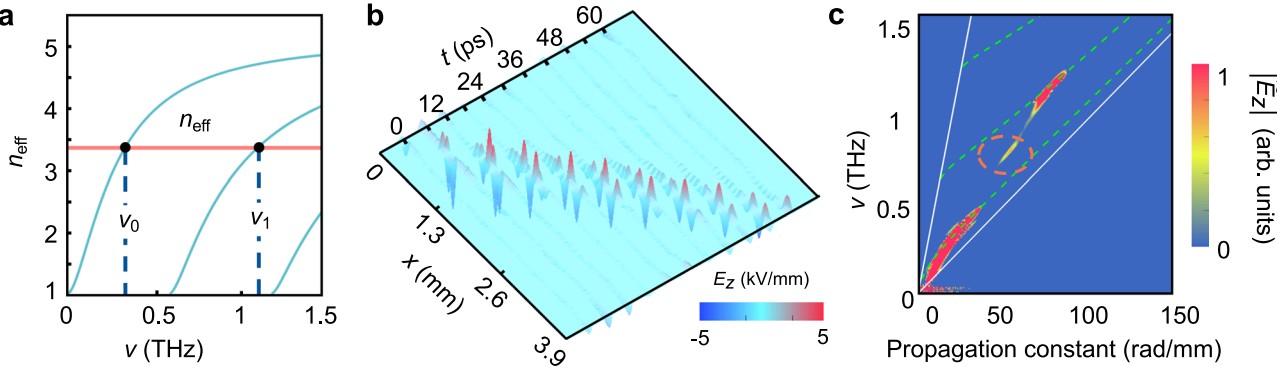

**Fig. 2 Experimental results of nonlinear generation of THz waves. a** Selection of velocity matching, where the zero and the first order TE waveguide modes of the THz waves plotted by cyan curves are chosen to match the pump laser pulses plotted as the red line. **b** Experimental observations of the spatiotemporal propagation of THz waves in LN waveguide. **c** The dispersion relation of THz waves by performing a 2D fast Fourier transformation of **b**. The two white solid lines indicate the light cone in vacuum and bulk LN, the dashed green curves represent the theoretical calculation for the waveguide modes, and the color mapping shows the THz field amplitude generated by the velocity matching technique, where the DFG signal is marked by a dashed red circle.

at 74° and the corresponding effective index is $n_{eff}$ tan$\alpha = 3.38$. Then, the zero and the first order waveguide modes are excited at their specific frequencies $v_0$ and $v_1$, respectively, as shown in Fig. 2a (higher-order modes are beyond the optical rectification envelop of pump laser). As both modes travel at the same phase velocity but with different frequencies ($v_0$ and $v_1$), their difference frequency can be generated by the nonlinear response of the medium.

The propagation of THz waves inside the LN waveguide causes a refractive index change in the LN wafer via the electro-optic effect (see in Methods)[42,46,47]. The probe beam is frequency-doubled to 400 nm and expanded to illuminate the entire sample. Then the probe beam is modulated after propagating through the LN wafer, and a corresponding phase shift is imparted proportional to the distribution of refractive index[42]. The phase-to-intensity conversion is accomplished by a phase-contrast imaging technique, so that the information could be collected by the CCD camera. The time delay between the pump and probe pulses is changed through moving the mechanical delay line, then the full spatiotemporal evolution of THz waves is obtained from the image sequence. Figure 2b shows the spatiotemporal propagation of the THz waves. By performing a 2D fast Fourier transformation to the spatiotemporal propagation function of the THz waves, the dispersion relation of the THz waves can be obtained, as shown in Fig. 2c. The THz wave profile (color mappings) matches well with the zero and first-order transverse electric (TE) modes of the LN waveguide (dashed green curves). Most importantly, the DFG signal is also observed, marked by a dashed red circle in Fig. 2c, which does not match any modes of the waveguide.

By choosing some characteristic snapshots from recorded Supplementary Movie 1, the entire process of the DFG can be directly visualized, as presented in Fig. 3, where Fig. 3a–d show the propagation patterns at 11, 22, 33, and 44 ps after the initial generation of the THz wave. In this system, the THz wave is not generated immediately after the pump laser pulse enters the sample, but it involves the entire process that lasts for about 10 ps (see in Supplementary Movie 1). During this process, the later-generated THz waves are superposed onto the former generated ones as they propagate, forming multi-cycle and quasi-monochromatic THz pulses[44]. The interaction length of the tilted-wavefront pump on the LN is estimated to be 0.89 mm, the distance that THz waves propagate in about 10 ps. Figure 3a, taken at 11 ps, presents the initial wave packet launched into the waveguide after the generation, where the two components with different frequencies are mixed in the same position. Then the two components gradually walk off due to the strong waveguide dispersion. Figure 3b, c shows the spreading of the wave-packet during propagation, which finally leads to an obvious separation between a faster low-frequency (LF) component and a slower high-frequency (HF) component. During this process, the phase-matching condition is satisfied because they both match with the pump laser, so the two components with central frequencies of $v_0$ and $v_1$ have identical phase velocity but different group velocities caused by the waveguide dispersion[45]. The identical phase velocity entails the observation and monitoring of the DFG signal, while the different group velocities lead to their walk-off in time, as shown in Fig. 3.

**Calculation and experimental estimation of the giant second-nonlinear susceptibility.** For further analysis, the electric field oscillations at four positions are presented in Fig. 4 as a function of time. Figure 4a presents the time-dependent signals obtained at the position of about $x_1 = 1.18$ mm, $x_2 = 1.82$ mm, $x_3 = 2.39$ mm, and $x_4 = 3.44$ mm. The coordinates of the positions are determined by setting $x = 0$ at the location where the generation of the THz waves is initially started. Figure 4b shows the pertinent Fourier transforms that reveal the spectral composition of the time signals. It can be seen that three frequency components essentially make up the total spectrum. When the THz waves reach position $x_1$, the two dominant peaks can be observed: one centered around the frequency $v_0 \approx 0.35$ THz belonging to the zero-order mode, and the other centered around the frequency $v_1 \approx 1.1$ THz pertinent to the first-order mode. Furthermore, there emerges a peak in the spectrum centered at the difference frequency $v_d = v_1 - v_0 \approx 0.76$ THz. This peak is of particular interest because it is the DFG signal of $v_0$ and $v_1$, which originates from the giant nonlinearity of LN. Here, the walk-off effect of the wave packets with central frequencies of $v_0$ and $v_1$ appears, since they have different group velocities due to the strong waveguide dispersion[45]. Therefore, at position $x_2 = 1.82$ mm, the DFG mode decays significantly and exhibits an obvious high radiation loss. At position $x_3 = 2.39$ mm, we can no longer see the DFG signal. In addition, all the modes are attenuated gradually with the propagation due to the frequency-dependent absorption in LN[48]. The DFG signal exhibits clearly a stronger dissipation because it is not supported by any modes of the waveguide. Additionally, the peak intensity of THz waves corresponding to frequency $v_0$ does not follow the same trend of $v_1$. Specifically, $v_1$ decreases only, whereas $v_0$ initially increases significantly and

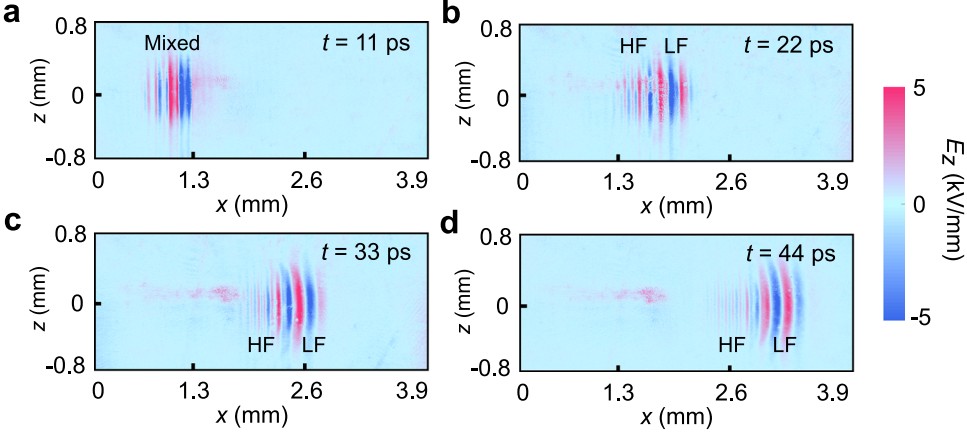

**Fig. 3 Direct observation of the DFG process for THz waves. a–d** are snapshots from Supplementary Movie 1 taken at 11, 22, 33, and 44 ps after the THz waves were initially generated. LF and HF indicate the lower and higher frequency components, and their walk-off effect during propagation is clear.

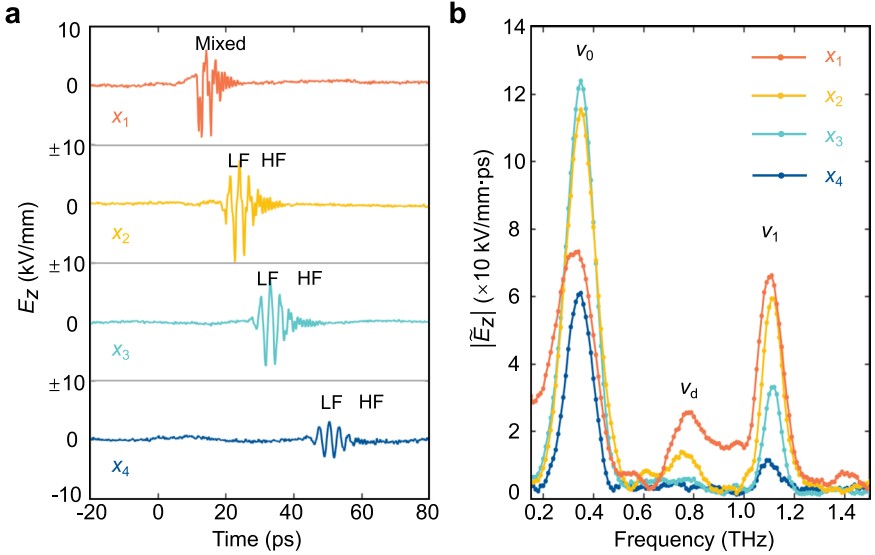

**Fig. 4 Evolution of the THz optical field with time at different positions and their corresponding Fourier spectra. a** The field oscillation of THz waves as a function of time at various positions of $x_1 = 1.18$ mm, $x_1 = 1.82$ mm, $x_1 = 2.39$ mm and $x_1 = 3.44$ mm relative to the position where THz waves are generated. LF and HF indicate the lower and higher frequency components which also show their walk-off during propagation. **b** Corresponding Fourier spectra of the matched THz waves at frequencies $v_0$ and $v_1$, as well as their DFG signal at the four positions.

only starts to decrease between positions $x_3$ and $x_4$. This is due to that the material absorption is much higher for $v_1$ than $v_0$[48], and also the generation power of $v_0$ and $v_1$ is different, so the peak field intensity for $v_1$ is much lower and also comes earlier than $v_0$. The DFG power should be proportional to the product of $v_0$ and $v_1$, but it also suffers from material absorption and large waveguide attenuation. All of the factors account for a very complicated intensity variation of the THz waves, as shown in Fig. 4b.

The spectrum at position $x_1 = 1.18$ mm is used to evaluate the nonlinear susceptibility in the experiment. As shown in Fig. 5a, each frequency component is fitted to a Gaussian function. Then, the electric field oscillation in the time domain of each frequency component is given by its inverse Fourier transform, as shown in Fig. 5b. The peak values of the electric fields for the three frequencies are $A_0(v_0) = 3.289 \times 10^6$ V/m, $A_1(v_1) = 1.713 \times 10^6$ V/m, and $A_d(v_d) = 0.935 \times 10^6$ V/m. These peak fields are chosen to make sure the evaluated value be a *lower limit* of the real nonlinear susceptibility, since it is difficult to choose real field intensities for the THz short pulses. Actually, it is nontrivial to calculate the second-order nonlinear susceptibility here as compared to that for

the absorption-free bulk materials, because of the tricky waveguide dispersion and the strong frequency-dependent absorption of the LN material. In order to estimate the experimental value of the second-order nonlinear susceptibility $\chi^{(2)}$, a custom model according to nonlinear optics was constructed which contains the pump laser generation of input THz waves with frequencies of $v_0$ and $v_1$, the mode attenuation of DFG frequency $v_1$, and the absorption of LN materials. In the model, we considered two special aspects in our system: (i) The mode dispersion of the LN waveguide[45] causes the major difference. On the one hand, the THz waves generated by the difference frequency $v_d = v_1 - v_0$ are not the eigenmodes, so they are not supported by the LN waveguide. On the other hand, the mode dispersion also causes a walk-off effect in time between THz waves with frequencies $v_0$ and $v_1$; (ii) The absorption of THz waves by the LN material also brings about some influence. Hence only the final result of $|\chi^{(2)}| > 1.58 \times 10^{-6}$ m/V is shown in Table 1, while the complete calculation is given in the Supplementary Note 1. To be consistent, all the employed evaluations and approximations are so taken to make sure that the calculated nonlinear susceptibility shows a lower limit of the real

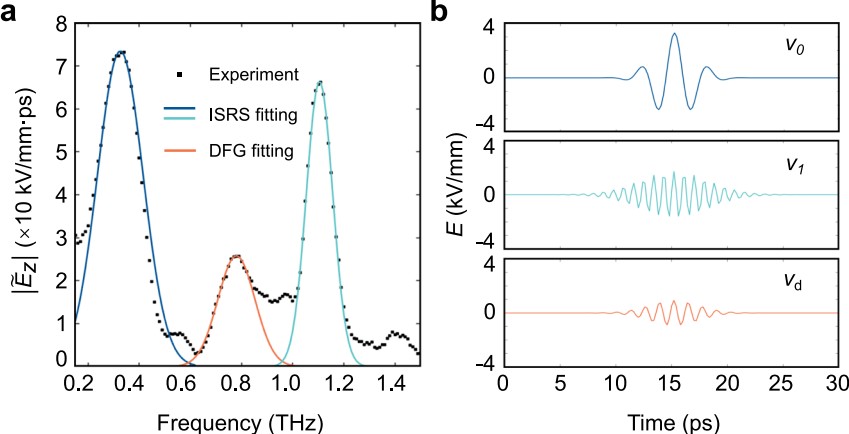

**Fig. 5 Fitting of measured DFG spectrum and comparison of second-order nonlinear susceptibility. a** Spectrum of the THz waves at position $x_1$ and Gaussian fittings of all the three frequency components. Blue and green curves fit $v_0$ and $v_1$, while the red curve fits the DFG signal $v_d$. **b** Corresponding inverse Fourier transform showing time domain oscillations for $v_0$, $v_1$, and $v_d$.

**Table 1 Comparison between experimentally obtained second-order nonlinear susceptibility and theoretically calculated values, along with prior results obtained at other frequency[36].**

| Frequency | $\chi^{(2)}$ (m/V, LN, extraordinary light) | | |
|---|---|---|---|
| | Theory[a] (ions) | Experiment[b] | Theory[a] (phonon polariton) |
| Visible light | $4.0 \times 10^{-11}$ | | |
| THz wave | $8.0 \times 10^{-11}$ | $>1.58 \times 10^{-6}$ | $\sim 1.7 \times 10^{-5}$ |
| Microwave | $6.7 \times 10^{-9}$ | | |

[a] Theoretical value (ions) is calculated by Lorentz model of ions, and theoretical value (phonon polariton) is calculated by nonlinear Huang equations, Eq. (1).
[b] Experimental values for visible light and microwave are quoted from[36].

value. Even so, the calculated value of $\chi^{(2)}$ at THz frequency is at least five orders of magnitude larger than that for visible light and three orders of magnitude larger than that for microwave radiation. Generally, the second-order susceptibility is a tensorial quantity. In our experimental setup, considering that the input and output THz signals are all polarized along the z-direction, the notation "$\chi^{(2)}$" here represents the tensorial element $\chi^{(2)}_{333}$ of LiNbO$_3$ crystal. Besides, additional results about pump-power dependence are provided and the pertinent results agree well with our expectation (see the Supplementary Note 2). Moreover, this result also shows remarkable advantages when compared with other common nonlinear optical materials, as seen in Supplementary Table 2. From Supplementary Table 2, it can be seen that the LiNbO$_3$ crystal has the largest phonon-polaritons enhanced nonlinear susceptibility, many orders of magnitude larger than those of traditional semiconductor crystals, organic crystals, and metasurfaces.

**Theroretical anaylsis of phonon-polariton enhanced non-linearity with modified Huang equations.** In order to give a clear explanation of the above observed nonlinear phenomenon, we employ classical nonlinear optics as the first principle. For the nonlinearity of electrons, the quantitative descriptions can be found in many textbooks[1,2]. The nonlinearity of ions can also be described in a similar way[49], and a value without the contribution of phonon polaritons is easy to calculate. When the contribution of phonon polariton is involved under an external driving of the

THz wave, the Lorentz oscillation model for lattices should be rewritten according to the Huang equations[37–39], shown as follows

$$\ddot{x} + \gamma\dot{x} + \omega_0^2 x = b_{12}E - ax^2 + (q/m)E_T(t), \quad (1a)$$

$$P = \epsilon_0(\epsilon_{(\infty)} - 1)E + b_{21}x. \quad (1b)$$

where the motion amplitude of ions is represented by $x$, with its electric charge and effective mass indicated by $q$ and $m$, and $E_T(t)$ stands for the corresponding driving THz field. The strength of the damping force and the anharmonicity (i.e., the nonlinearity) are characterized by parameters $\gamma$ and $a$, respectively. Here $\omega_0$ represents the eigen angular frequency of the ions, and $\epsilon_0$ is the permittivity of vacuum. In particular, the term $b_{12}E$ in Eq. (1a) represents the contribution of phonon polariton, where the values of the coupling coefficients are determined by $b_{12} = b_{21} = \omega_0\sqrt{\epsilon_0(\epsilon_{(0)} - \epsilon_{(\infty)})}$, with $\epsilon_{(0)}$ and $\epsilon_{(\infty)}$ stand for the low- and high-frequency relative permittivity of the material, respectively[37–39]. $E$ shows the macroscopic electric field radiated by the ions. Equation (1b) describes the dependence of the polarization $P$ on $E$ field and $x$.

With the help of Maxwell's equations, the polarization $P$ can be eliminated from the above equations, and thus Eq. (1a, 1b) can be written as

$$-\omega^2 x - j\gamma\omega x + \omega_0^2 x = b_{12}E - ax^2 + (q/m)E_T(t),$$
$$-\epsilon_0\mu_0\omega^2\epsilon_{(\infty)}E - \omega^2\mu_0 b_{21}x - \nabla^2 E = 0 \quad (2)$$

Here, the conventional method in nonlinear optics is used[1], and it is assumed that $x = x_0 e^{-j\omega t}$ and $E = E_0 e^{-j\omega t}$ with $\omega$ being the angular frequency of the driving THz field, where $x_0$ and $E_0$ are independent of $t$. The permeability of the vacuum is taken to be $\mu_0 = 4\pi \times 10^{-7}$ H/m when a non-magnetic material is considered. If the propagation of $E$ in space is described as $e^{j\sqrt{\epsilon_{(0)}}kr}$, the derivative of $E$ to space can be eliminated as $\nabla^2 E = -\epsilon_{(0)}k^2 E = -\epsilon_{(0)}(\omega^2/c^2)x$. Then, the following equation can be obtained

$$\frac{\epsilon_{(0)} - \epsilon_{(\infty)}}{c^2}E = \mu_0 b_{21}x. \quad (3)$$

Here, $c = 1/\sqrt{\epsilon_0\mu_0}$ indicates the light speed in the vacuum. From Eq. (3), it can be seen that $b_{12}E = \omega_0^2 x$. Substituting it into

Eq. (2), the restoring force is canceled, as shown in Eq. (4).

$$-\omega^2 x - j\gamma\omega x = -ax^2 + \left(\frac{q}{m}\right)E_T(t). \qquad (4)$$

This equation is known as the Drude dispersion equation, which indicates a remarkable delocalization of phonon-polaritons, since the restoring force is canceled from the Lorentz dispersion equation. The mobility of phonon-polaritons is comparable to that of the free electrons in a metal, which is described by the classical Drude dispersion model. This also suggests a theoretical framework to show why the obvious propagation of phonon-polaritons was experimentally observed[40,45]. By adding the phonon polaritons, the ionic nonlinearity can be significantly improved, which is very important for the remarkable enhancement of the nonlinearity at THz frequencies, as observed in our experiment.

The second-order nonlinear susceptibility can be calculated by the Miller's rule[46,50], and the Miller's constant is given by

$$\delta_M = \frac{\chi^{(2)}(\omega_3; \omega_1, \omega_2)}{\chi^{(1)}(\omega_1)\chi^{(1)}(\omega_2)\chi^{(1)}(\omega_3)}, \qquad (5)$$

where the linear susceptibilities $\chi^{(1)}$ can be calculated from Eq. (1a, 1b). By using the definition $D(\omega_i) = -\omega_i^2 - j\omega_i\gamma$, the linear susceptibility can be written as[1]

$$\chi^{(1)}(\omega_i) = \frac{Nq^2}{\epsilon_0 m}\frac{1}{D(\omega_i)}. \qquad (6)$$

Here $i = 1,2,3$, $j$ is the imaginary unit, $N$ is the number of oscillators per unit volume, and $\epsilon_0 = 8.85 \times 10^{-12}$ F/m. In Eq. (5), the magnitude of the Miller's constant is determined by

$$|\delta_M| = \frac{ma\epsilon_0^2}{N^2 q^3}. \qquad (7)$$

Under the assumption that the magnitudes of the harmonic and anharmonic terms in the potential energy should be equal when $x$ is of the same order of magnitude as the lattice spacing[1,46], we can get the parameter

$$a \approx \omega_0^2 N^{1/3}. \qquad (8)$$

Take the values for an LN crystal, $\omega_0 = \omega_{TO} = 7.6$ THz, which is the resonant frequency of the lowest order of transverse optical phonons. The values of the other parameters[51,52] are $N = 6.29 \times 10^{27}$ m$^{-3}$ and $\gamma = 0.84$ THz. The transverse optical phonons are considered as an $A_1$ soft mode[18], $m_+ = m(Nb^{5+}) = 93$ u, and $m_- = 3m(O^{2-}) + m(Li^+) = 55$ u, where 1 u $= 1.66 \times 10^{-27}$ kg. The reduced mass is $m = (m_+ m_-)/(m_+ + m_-)$, and the charge $q = 5e$, where $e = 1.60 \times 10^{-19}$ C is the elementary charge. The second-order nonlinear susceptibility at THz frequency is calculated to be $|\chi_{pp}^{(2)}| = 1.70 \times 10^{-5}$ m/V from Eqs. (4) to (8). In contrast, if only the ionic nonlinearity is considered but neglecting the phonon polaritons, the value can be simply calculated by the Lorentz model for the ions, which leads to $|\chi_{ion}^{(2)}| = 8.0 \times 10^{-11}$ m/V. Clearly, our experimental result is much closer to $|\chi_{pp}^{(2)}|$ but orders of magnitude higher than $|\chi_{ion}^{(2)}|$, indicating that the phonon polariton does contribute and plays the major role for the observed THz nonlinearity.

We have presented a direct observation of the dynamic process of THz DFG in LN wafer, where the nonlinear susceptibility is found to be orders of magnitude larger than that in visible and microwave regimes. In our experiment, the signal from the DFG is selectively enhanced (compared with that from the SHG), and the corresponding nonlinear susceptibility was calculated based on the assumption that the effect of pump laser is totally ignored (see Supplementary Note 2 for details). This unusual phenomenon is explained by introducing a nonlinear modification to the classical Huang equations, providing also a theoretical explanation to the delocalization of phonon polaritons for the first time to the best of our knowledge. In the ionic crystal, the delocalized phonon polaritons can lead to a giant enhancement of the optical nonlinearity at THz frequency by increasing the ionic polarization. Although in our experiment, both the input and output THz signals are generated inside the crystal, this giant nonlinearity mediated by phonon polaritons still works when the THz waves are inputted from external sources, just as in quantum cascade lasers. Furthermore, using better monochromatic THz sources would obtain a higher efficiency because then the interaction distance could be much longer. Meanwhile, the system loss could also be reduced in a subwavelength waveguide, by a better waveguide mode design, or a better growing crystal. Such high nonlinearities may find valuable practical applications such as on-chip integration of THz waves. For example, the giant nonlinearity at THz frequency induced by phonon polaritons may be employed for the generation of THz supercontinuum spectra or THz frequency combs, which may find applications in numerous physical, chemical, and biological systems with characteristic THz fingerprints. In addition, the nonlinear susceptibility for high-frequency light can also be enhanced by delocalized phonon polaritons, because electrons could inherit immense nonlinearity from the delocalized phonon polaritons by coupling with them, which is useful for many applications including optical control of spin qubits in semiconductors. Moreover, the delocalized phonon polaritons may enable effective modulations to many properties of ionic crystals (such as in optomechanics, thermo-optics, electro-optics, and magneto-optics), as well as domain structures and phase transitions of ferroelectric/ferromagnetic crystals.

## Methods

**Generation of THz waves**. The experiment is performed using a Ti: sapphire femtosecond laser system with a central wavelength of 800 nm, a repetition rate of 1 kHz, and a pulse duration of about 120 fs. The laser output is split into a pump pulse (450 μJ, beam radius of about 0.5 mm) and a probe pulse (frequency-doubled, 400 nm, 50 μJ, beam radius of about 8 mm), and they can be delayed relatively to one another to give a time resolution of about 100 fs for the time-resolved imaging system.

To generate two THz waves with different frequencies simultaneously, a grating with 1200 lines/mm is used to tilt the wavefront of the pump pulse, thus to match the first two orders of the waveguide modes. The grating is imaged at the entrance facet of the sample by directing the first-order diffraction through a single cylindrical lens with a focal length of 10 cm[44].

The tilt angle $\alpha$ of the pulse on the sample surface depends on the angle of incidence of the pump beam onto the grating as well as on the magnification $M$ of the imaging system. The angle of incidence is held constantly at 20°, which yields a tilt angle $\alpha_1 = 46°$ for the first-order diffraction light. The magnification of the one-lens imaging system is varied by moving the grating and lens while keeping the location of the image plane fixed on the sample surface. The resulting tilt angle at the sample is given by the relation $\tan\alpha = \tan\alpha_1/M$. The interaction length, determined by the pump spot size on the grating as well as the magnification, can achieve a very broad range[44]. To generate the THz waves, a magnification $M = 0.31$ is chosen here, which yields $\alpha = 74°$ and an effective refractive index of $n_{eff} = 3.38$.

**Detection of THz waves**. Generated THz wave propagates along the $x$ direction. To visualize the THz field, a probe beam is used that is spatially filtered and expanded to illuminate the whole sample homogeneously[42,51]. Since LN is an electro-optical material, the THz wave would cause a refractive index variation $\Delta n$ $(x,z)$ when propagating in the waveguide. The probe beam propagates in $y$ direction through the sample of thickness $l$ and thereby gathered an integral phase-shift $\Delta\phi_{opt}(x,z)$ in respect to the refractive index variation. The relation between the THz field $E_{THz}(x,z)$ and the phase shift is given by[42]

$$\Delta\phi_{opt}(x,z) = 2\pi\frac{l}{\lambda_{opt}}\Delta n(x,z) = 2\pi\frac{l}{\lambda_{opt}}\frac{n_e^3 r_{33}}{2}E_{THz}(x,z) \qquad (9)$$

where $l$ is the thickness of the sample and $\lambda_{opt}$ is the wavelength of the probe beam. Extraordinary polarization is chosen for the pump, probe and THz waves, to achieve an optimum THz signal. Accordingly, $n_e$ and $r_{33}$ in Eq. (9) refer to the (extraordinary) refractive index and electro-optic coefficient of the LN crystal, respectively.

In order to visualize the phase-shift, a phase-contrast method is employed. This method involves imaging the sample onto a CCD camera using a 4f system with two achromatic lenses having a focal length of 10 cm and usage of a $\lambda/4$ phase plate in the Fourier plane (focal plane) of the system. The phase plate introduces a $\lambda/4$ phase shift to the zero-order diffraction component in the Fourier plane. Thus, the phase image is transformed into an intensity image with[42,47,52]

$$I(x,z) = I_0(x,z)\left\{3 - 2\left[\cos\Delta\phi_{\mathrm{opt}}(x,z) - \sin\Delta\phi_{\mathrm{opt}}(x,z)\right]\right\}$$
$$= I_0(x,z)\left[3 - 2\sqrt{2}\cos\left(\frac{\pi}{4} + \Delta\phi\right)\right] \qquad (10)$$

where $I_0(x,z)$ is the intensity distribution of the original probe beam (at $\Delta\phi_{\mathrm{opt}} = 0$). Here, the field intensity of THz waves is very strong, so the specific value of Eq. (10) cannot be treated as a linear approxiamation[40]. Consequently, the phase in the sample can be obtained from

$$\Delta\phi_{\mathrm{opt}}(x,z) = \arccos\left[\frac{\sqrt{2}}{4}\left(3 - \frac{I}{I_0}\right)\right] - \frac{\pi}{4}. \qquad (11)$$

Therefore, the electric field intensity of the THz waves could be calculated according to the measurement of the probe intensity with and without THz waves.

## Data availability

The data that support the findings of this study are available from Y.L. or Q.W. upon reasonable request.

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

## Acknowledgements

This work was funded by the National Natural Science Foundation of China (11974192, 61705013), the National Key R&D Program of China (2017YFA0303800), the Foundation of State Key Laboratory of Laser Interaction with Matter (SKLLIM1903), the 111 Project (B07013), and the Program for Changjiang Scholars and Innovative Research Team in University (IRT_13R29). We appreciate the helpful discussion with Prof. Weiping Zang and the constructive advice from Profs. Roberto Morandotti, Romano A. Rupp, and Dr. Alessandro Tomasino.

## Author contributions

Q.W. and J.X. provided the basic idea, Y.L., Q.Z. and Q.W. developed the concept, Y.L. and Q.Z. implanted the experiment and carried out the data analysis with the help of Q.W. and Z.C., Y.L., Q.W., Z.C., Q.Z., and X.L. wrote the paper, Q.W. and J.X. supervised the project.

## Competing interests

The authors declare no competing interests.
