## [Peer Review File · Nature Communications]

Editorial Note: Parts of this peer review file have been redacted as indicated to remove third-party material where no permission to publish could be obtained.

REVIEWER COMMENTS

Reviewer #1 (Remarks to the Author):

The manuscript by Lu et al. reports the novel observation of a large THz nonlinearity in a LiNbO₃ waveguide, which the authors attribute to an enhancement effect due to phonon-polaritons. This effect is potentially interesting for THz optics application and the manuscript seems suited for Nature Communications.

In these experiments, mostly following a THz generation technique developed in Ref. 42, narrowband THz pulses are generated into the LiNbO₃ waveguide and imaged through the phase shift of a 400nm probe beam. The spectral content of the THz spectrum shows two main THz peaks and a difference frequency spectral component which disappears for increasing walk-off of the two THz colors in the waveguide. The key finding of the paper, i.e. the giant second order susceptibility is determined from a fit of the spectrum in Fig. 5.

I mainly have two technical comments which deserve some clarification:

1) This study relies on an absolute measurement of the THz field. Right now it is unclear how robust is the calibration between phase contrast and field amplitude and the authors should discuss possible sources of uncertainty. Furthermore, for very strong fields one might have cases of overrotation of the imaging phase, in a way similar to overrotation of the electro-optic polarization in the electrooptic sampling. How can the authors make sure that their detection is still in the linear regime and the distorted waveform in Fig. 4a X1 is not affected by this phenomenon?

2) the second order susceptibility is a tensorial quantity, but this paper discusses it as if LiNbO₃ was an isotropic medium. Could the authors specify which tensorial element of the optical susceptibility is affected by the phonon polariton enhancement?

Reviewer #2 (Remarks to the Author):

The authors used tilted-wavefront ultrafast pulses to generate THz phonon-polaritons in a slab waveguide made of lithium niobate (LN). And they utilized the phase-contrast imaging technique to image the propagation of phonon-polaritons in time domain. In addition of two waveguide modes at two different central frequencies, they observed a frequency corresponding to the difference between the two frequencies of the waveguide modes. The authors claimed this is difference frequency generation (DFG) from the two waveguide modes at THz frequency regime, and the second order susceptibility $\chi^{(2)}$ is order of magnitude higher than that in the visible and microwave regimes.

The most interesting part is the observation of the difference frequency in addition to the two waveguide modes. However, there is potential problem to claim giant $\chi^{(2)}$ on the order of 10^{-6} at THz frequency. The authors only considered DFG from the two phonon-polariton modes at THz frequency, leading to such a giant $\chi^{(2)}$. However, they neglected that the pump laser also can directly generate this difference frequency through the impulsive stimulated Raman scattering (ISRS), just like the other two frequency modes. If ISRS dominates the generation mechanism of this difference frequency, the calculated $\chi^{(2)}$ in the present manuscript is totally misleading. Several issues as follow should be addressed.

1. What is the interaction length of the tilted-wavefront pump on the LN? Can the spot of the pump on the LN be imaged by the CCD camera?

2. In Eq. (S3-2), it is strange to use an exponential decay function to describe the effective intensity of the pump laser. Shouldn't the pump spot be similar to Gaussian like function (rise-peak-decrease) rather than monotonically decay function? What value of Gamma was used for the fitting results?
3. In Eq. (S3-3), ISRS should also be considered. E_d should not be only from the amplitudes of the two waveguide frequency modes.
4. What exact function was the applied driving THz field $E_T(t)$ taken for the calculation through Eq. (4)-Eq.(8) to obtain the theoretical $\chi^{(2)}_{pp} = 1.7 \times 10^{-5} \text{ m/V}$? Does the calculated $\chi^{(2)}$ depend on different amplitudes of the applied driving THz field?
5. Pump-power dependent measurement should be helpful to clarify the generation mechanism of the difference frequency. The relation between the intensity of the difference frequency and the pump power should be different for generation through ISRS and generation purely from two different waveguide modes. The theory proposed in this manuscript can also be verified by pump-power dependent measurement.

Reviewer #3 (Remarks to the Author):

In this work Lu et al. demonstrate the difference frequency generation (DFG) of two THz waves generated with a femtosecond pump laser through the impulsive stimulated Raman scattering process in a Lithium Niobate (LN) waveguide. They estimate a giant 2nd order nonlinear susceptibility, χ_2 , in the order of 10^{-6} m/V , which is attributed to phonon-polaritons in the LN crystal. While there are previous works on THz wave generation (Ref 42) and DFG of near infrared input signals (Sasaki & Yuri, Appl. Phys. Lett. 81, 3323, 2002) in LN, as far as I know this is the first observation of DFG where both the input and output signals are in the THz range. I found the experiment very interesting, the evidence for the DFG signal convincing, and the time-resolved imaging of the THz waves in the video beautiful. The giant value of χ_2 makes LN a promising material for frequency conversion of THz signals, which is useful for many applications including optical control of spin qubits in semiconductors. However, I think there are many important issues that need to be addressed before the paper is suitable for publication in Nature Communication.

1. The estimation of the experimental χ_2 can be presented in a more transparent way. In the Materials and Methods χ_2 is stated to be obtained from "solving Eq. S3-2 and S3-3." Given the importance of χ_2 in this work, I think the authors should express clearly how χ_2 depends on the experimental and fitting parameters. This does not seem hard to do, since Eq S3-2 has an analytical solution. A more detailed description of how the experimental χ_2 is estimated should also be provided in the main text.

2. There are three important fitting parameters used in the estimation of the experimental χ_2 : The effective intensities I_{G0} , I_{G1} , and the decay constant Gamma, introduced in Eq. S3-2 for explaining why the lower-frequency THz wave (ν_0) at first increases with the distance of propagation up to $x_3 = 2.39 \text{ nm}$. I could not find the fitted values of these parameters in the paper. These values should be stated along with a justification for why they are reasonable and consistent with the underlying physics of the impulsive stimulated Raman scattering process. Moreover, what is the value of the waveguide mode attenuation M in Eq. S3-2? Is it another fitting parameter?

3. The authors claimed that the value of χ_2 in their work is many orders of magnitude larger than those obtained for visible light and microwave but did not cite any references. Their claim could be strengthened by a table of comparison between their χ_2 and those reported for other materials in the literature. I am not sure why the authors compare their result with only visible and microwave values. A comparison with χ_2 of other materials in the THz and infrared ranges is more relevant.

4. The significance of the result, especially its relevance for THz applications, should be given more discussion. It is stated in the paper that the giant χ^2 of LN can be useful for THz technology. Unlike in this experiment where both the input and output THz signals are generated inside the crystal, in most frequency conversion applications the input signals are generated from an external source, for example a quantum cascade laser. Is it possible to design a LN waveguide for DFG and SFG (sum-frequency generation) of external inputs in the THz range? In this experiment strong loss is observed for the output DFG signal, which may limit its uses. Is there a way to reduce the loss?

5. For completeness previous works on THz DFG in LN, for example Sasaki & Yuri, Appl. Phys. Lett. 81, 3323, 2002, should be discussed. It would be great for researchers in the field if the authors can make clear what the advances are in their work compared with previous works on THz DFG.

I also have a few minor concerns and questions below:

6. In the analysis the authors used a formula for monochromatic inputs for E_{DFG} in Eq. S3-3. However, it is shown in Fig. 4b that the input THz waves are pulses with a significant frequency spread of about 0.1 THz. In this case E_{DFG} is given by an integral over the different frequency components of the inputs. The effect of this spectral width may be significant, as it is comparable to the central frequency of the lower-frequency THz wave. Can the authors comment on why the monochromatic approximation is valid for the estimation of the experimental χ^2 ?

7. In the third paragraph of the Introduction it is stated that the strong nonlinearity in Refs 26-34 is due to "high mobility of the electrons." Mobility is defined for charge carriers. In some of the cited works, for example Refs 26 and 28, strong nonlinearity is observed for electrons in bound states where mobility is not a defined concept. The strong nonlinearity in these systems comes from the large extent of the wavefunction resulting in a huge dipole coupling with fields.

8. What are the beam radii of the pump and probe femtosecond lasers? These are necessary for estimating the intensities and the electric fields of the pump and the probe.

9. The notation M is used for both the magnification of the imaging system and the waveguide mode attenuation.

RESPONSE TO THE REVIEWER COMMENTS

Reviewer #1 (Remarks to the Author):

The manuscript by Lu et al. reports the novel observation of a large THz nonlinearity in a LiNbO₃ waveguide, which the authors attribute to an enhancement effect due to phonon-polaritons. This effect is potentially interesting for THz optics application and the manuscript seems suited for Nature Communications.

In these experiments, mostly following a THz generation technique developed in Ref. 42, narrowband THz pulses are generated into the LiNbO₃ waveguide and imaged through the phase shift of a 400nm probe beam. The spectral content of the THz spectrum shows two main THz peaks and a difference frequency spectral component which disappears for increasing walk-off of the two THz colors in the waveguide. The key finding of the paper, i.e. the giant second order susceptibility is determined from a fit of the spectrum in Fig. 5.

Response0: Thank the reviewer for the positive comments on our work. As the reviewer pointed out clearly, the second-order nonlinearity at THz frequency is potentially important for the THz optics.

I mainly have two technical comments which deserve some clarification:

1) This study relies on an absolute measurement of the THz field. Right now, it is unclear how robust is the calibration between phase contrast and field amplitude and the authors should discuss possible sources of uncertainty. Furthermore, for very strong fields one might have cases of overrotation of the imaging phase, in a way similar to overrotation of the electro-optic polarization in the electrooptic sampling. How can the authors make sure that their detection is still in the linear regime and the distorted waveform in Fig. 4a X1 is not affected by this phenomenon?

Response1: We thank the reviewer very much for this comment and the very helpful advice. Here the reviewer mainly has two concerns about the quantitative measurement of the THz waves generated in the LiNbO₃ waveguide. One is the robustness of the relation between THz field amplitude and the phase contrast, and the other is whether the THz amplitude in our experiment exceeds the linear regime of the electrooptic sampling.

This detection relation is given in Methods 2, as Eq. (9) in the main text (Page 9 Line 304) shows, also in Ref. 41 [Q Wu, et al., Opt. Express 17, 9219 (2009)], which reads

$$\Delta\phi_{\text{opt}}(x, z) = 2\pi \frac{l}{\lambda_{\text{opt}}} \Delta n(x, z) = 2\pi \frac{l}{\lambda_{\text{opt}}} \frac{n_e^3 r_{33}}{2} E_{\text{THz}}(x, z)$$

and the phase shift $\Delta\phi_{\text{opt}}$ is experimentally measured by recording the probe distribution
 with and without THz waves, shown in following Eq. (R1) [Ref. 41, Q Wu, et al., Opt.
 Express 17, 9219 (2009)]

$$\Delta\phi_{\text{opt}}(x, z) \approx \frac{1}{2} \left[\frac{I(x, z)}{I_0(x, z)} - 1 \right] = \frac{1}{2} \frac{\Delta I(x, z)}{I_0(x, z)} \quad (\text{R1})$$

According to the reviewer's comments, we carefully considered the possible sources of the
 uncertainty in this measurement, which mainly come from the fluctuation of the probe laser
 and the strong THz field intensity (should be in the linear regime as the reviewer's second
 concerns).

 1. As the reviewer mentioned, a large field amplitude would cause an over rotation of the
 imaging phase. The maximum field amplitude is at x_1 position in Fig. 4(a), about 6.89
 50 kV/mm. Thanks to the reviewer reminding us that this value would bring a large
 deviation, so we considered that and recalculated it in the revised manuscript. To
 evaluate the real deviation of the approximation in Eq. (9) in the main text, we could
 simply deduce the phase shift according to the practical THz field amplitude E_{THz} .
 According to Eq. (9) in the main text, the phase shift of the probe depends on the value
 of the THz fields in the experiment. According to principles in phase-contrast imaging
 [Ref. 52, E. Hecht, Optics, fifth edition (Pearson Education Limited, 2017), Global ed],
 the practical value of I/I_0 is

$$\{3 - 2[\cos \Delta\phi - \sin \Delta\phi]\} = 3 - 2\sqrt{2} \cos\left(\frac{\pi}{4} + \Delta\phi\right), \quad (\text{R2})$$

 while the linear approximation gives a measured value of I/I_0 of [Ref. 41, Q Wu, et al., Opt.
 Express 17, 9219 (2009)]

$$\{3 - 2[\cos \Delta\phi - \sin \Delta\phi]\} \approx [1 + 2\Delta\phi]. \quad (\text{R3})$$

In the linear approximation, according to Eq. (R1), the detected value E'_{THz} satisfies

$$\Delta\phi' = \frac{I - I_0}{2I_0} = 2\pi \frac{l}{\lambda_{\text{opt}}} \frac{n_e^3 r_{33}}{2} E'_{\text{THz}}. \quad (\text{R4})$$

Considering the practical value E_{THz} , it should be

$$\Delta\phi = \arccos\left[\frac{\sqrt{2}}{4} \left(3 - \frac{I}{I_0}\right)\right] - \frac{\pi}{4} = 2\pi \frac{l}{\lambda_{\text{opt}}} \frac{n_e^3 r_{33}}{2} E_{\text{THz}}. \quad (\text{R5})$$

Accordingly, the relation between E'_{THz} and E_{THz} can be obtained as follows:

$$E_{\text{THz}} = \frac{\frac{I - I_0}{2I_0}}{\arccos\left[\frac{\sqrt{2}}{4} \left(3 - \frac{I}{I_0}\right)\right] - \frac{\pi}{4}} E'_{\text{THz}} \quad (\text{R6})$$

Use E'_{THz} to substitute I/I_0 , we obtain that

$$E_{\text{THz}} = \frac{2\pi \frac{l}{\lambda_{\text{opt}}} \frac{n_e^3 r_{33}}{2} E'_{\text{THz}}}{\arccos \left[\frac{\sqrt{2}}{2} \left(1 - 2\pi \frac{l}{\lambda_{\text{opt}}} \frac{n_e^3 r_{33}}{2} E'_{\text{THz}} \right) \right] - \frac{\pi}{4}} E'_{\text{THz}}. \quad (\text{R7})$$

Here, we give a mapping from approximated value to the practical value [Eq. (R7)], as Fig.
 R1 shows.

 Fig. R1. The deviation of quantitative measurement as a function of THz field
 intensity. The blue solid line indicates the practical value, the blue dashed curve
 indicates the measurement, and the green curve represents the imaging phase. The
 maximum THz field is indicated with blue circles.

 When the measured THz wave is 6.89 kV/mm, the practical value is 8.78 kV/mm, here the
 maximum value of the relative deviation is $(8.78-6.89)/8.78=21.6\%$. In order to make the
 results accurate, the mapping of Fig. R1 was performed to the experimental data/results, and
 the relative figures and data [Fig. 4, Fig.5, and Fig. S1] have been modified accordingly. We
 have also rewritten Methods 2, and the linear approximation was removed to get a more
 accurate value (Page 9 Lines 313~318). Thank the reviewer again for these important and
 helpful comments.

 According to the new data, we calculated the electric field values of v_0 , v_1 , and v_d at the
 position x_1 are 3.289 kV/mm, 1.713 kV/mm, and 0.935 kV/mm, respectively. We also
 changed the model according to Reviewer 2 (Comment 2, Line 205). In the revised version,
 the new calculated $\chi^{(2)} = 1.58 \text{ mm/kV}$, which still agrees with the theoretical expectation.

 The other factor that may cause uncertainty is the fluctuation of the probe laser. Here, we
 measured the fluctuation of the laser pulse, as shown in Figure R2. Since a single data
 collection in our experiments takes about 80 min, we recorded the power fluctuation of the

laser pulse in 2 hours. From Fig. R2, the fluctuation of the laser power is calculated as about
0.83% (Source data is provided in Supplementary Materials). Since the probe is frequency-
doubled, the pertinent fluctuation is about $1-(1-0.83\%)^2$, which is 1.65%. According to the
dependency of the THz electric field and the probe, this error exerts a negligible influence on
the experiment results.

Fig. R2. The fluctuations of the laser output used in the experiment.

2) the second order susceptibility is a tensorial quantity, but this paper discusses it as if
LiNbO₃ was an isotropic medium. Could the authors specify which tensorial element of the
optical susceptibility is affected by the phonon polariton enhancement?

**Response2:** Thank the reviewer very much for this point. In our manuscript, the tensorial
element of the second-order susceptibility is not mentioned in the last version. According to
the experimental setup and the properties of the phonon polaritons, the input and output DFG
signals are all polarized along z -direction, so the tensorial element we studied in our work is
$\chi_{333}^{(2)}$ or d_{33} of LiNbO₃ crystal. From the theoretical consideration, the phonon polariton
should contribute to enhancing the nonlinearity in all the tensorial elements. However, the
strong anisotropy of the LiNbO₃ crystal makes the polarization of phonon polariton in xy
plane very weak, and thus it mainly works for $\chi_{333}^{(2)}$. But for other isotropic ionic crystal (such
as GaP, ZnTe), the phonon polariton can enhance all the tensorial elements. In addition,
according to the nonlinear modified Huang equations (Eq. (2), or the deduced Eq. (4) in the
main text), the contribution of the phonon polariton not only can enhance the second-order
susceptibility, but it can also enhance the higher-order nonlinear susceptibilities, which
makes it more significant in the future nonlinear technologies at THz frequency. In the
revised manuscript, we added a necessary explanation to clarify which tensorial element is
involved in our study.

Original Text (Page 5, Lines 189~191):

“Generally, the second-order susceptibility is a tensorial quantity. In our experimental setup,
considering that the input and output THz signals are all polarized along the z -direction, the
notation “ $\chi^{(2)}$ ” here represents the tensorial element $\chi_{333}^{(2)}$ of LiNbO₃ crystal.”

Reviewer #2 (Remarks to the Author):

The authors used tilted-wavefront ultrafast pulses to generate THz phonon-polaritons in a
slab waveguide made of lithium niobate (LN). And they utilized the phase-contrast imaging
technique to image the propagation of phonon-polaritons in time domain. In addition of two
waveguide modes at two different central frequencies, they observed a frequency
corresponding to the difference between the two frequencies of the waveguide modes. The
authors claimed this is difference frequency generation (DFG) from the two wave guide
modes at THz frequency regime, and the second order susceptibility $\chi^{(2)}$ is order of
magnitude higher than that in the visible and microwave regimes.

The most interesting part is the observation of the difference frequency in addition to the two
waveguide modes. However, there is potential problem to claim giant $\chi^{(2)}$ on the order of
10^{-6} at THz frequency. The authors only considered DFG from the two phonon-polariton
modes at THz frequency, leading to such a giant $\chi^{(2)}$. However, they neglected that the
pump laser also can directly generates this difference frequency through the impulsive
stimulated Raman scattering (ISRS), just like the other two frequency modes. If ISRS
dominates the generation mechanism of this difference frequency, the calculated $\chi^{(2)}$ in the
present manuscript is totally misleading. Several issues as follow should be addressed.

**Response0:** Thank the reviewer for the careful evaluation of our work. As the reviewer
mentioned, the DFG signal appears between the two waveguide modes, but it itself was not
supported by any waveguide mode. This is a powerful evidence to show that the difference
frequency signal is generated by the DFG of the two THz waveguide modes, rather than
generated by the velocity match of laser pulses via ISRS. The direct observation of such a
DFG process also gives a dynamic view of it.

The reviewer here also concerns that the pump laser could generate THz waves at difference
frequency, which may cause misleading of the reported nonlinear phenomenon. Here, we
elaborate on this point and exclude the influence of pump laser on the DFG signals from the
following aspects.

(1) THz waves generated by pump laser (ISRS) have a broadband spectrum, which is
centered at about 0.5 THz, seen in our previous work, [Ref. 41 (Q Wu, et al., Opt.
Express 17, 9219 (2009))]. However, in the tilted-wavefront generation, only the
frequency components that match the waveguide modes remain, as shown in Fig. 4(b),
where even the energy distribution at about 0.5 THz is still very low. If the pump
laser could generate THz waves at the difference frequency, it must also generate THz

waves at the frequency range between the two waveguide modes, especially at 0.5
162 THz, rather than only at the difference frequency point.

(2) Even if we assume ISRS could generate THz waves at the difference frequency point,
it is different from the other two frequencies that match with the waveguide modes,
where the generated THz waves could have a continual increase as a function of
position x . Considering the general generation efficiency [Ref. 40 (T. Feurer, et al.,
*Annu. Rev. Mater. Res.* 37, 317 (2007)), and Ref. 41 (Q Wu, et al., *Opt. Express* 17,
9219 (2009))], a 400 μ J pump laser beam could generate THz waves with a maximum
amplitude of 0.4 kV/mm focused by a cylindrical lens to a line of about 0.1 mm. Here
we consider an extreme condition that the wavefront-tilted pump laser excites THz
waves simultaneously in the LiNbO₃ waveguide, since the length of the pump is much
larger (about 0.9 mm, see the Response1 (Lines 190~199), the field amplitude of the
pump is about 1/9 of that in Ref. 40. Accordingly, the THz intensity it generated is
only 1/9² of that value [Ref. 40 (T. Feurer, et al., *Annu. Rev. Mater. Res.* 37, 317
(2007))], which is about 0.005 kV/mm. Considering our practical setup, where the
wavefront-tilted pump does not generate THz waves in a very short time span and for
a very large waveguide mode attenuation, the practical THz waves at difference
frequency point are far smaller than 0.005 kV/mm, which is negligible compared with
the DFG generated value of about 0.935 kV/mm (see Table S1 in the Supplementary
Materials Page 2 Line 57, or in the main text Page 5 Line 169).

(3) In addition, the ISRS only contributes when the pump laser was in the LiNbO₃
waveguide. This effect was excluded by just analyzing the experimental data after the
pump laser left the waveguide (Response 1, Lines 192~201).

According to the above discussions, we think the contribution of pump laser generation at
the DFG frequency is ignorable, and it makes little difference to our results. However, to
clarify this important issue, we have changed the presentation in both the main text and the
Supplementary Materials of our revised manuscript.

1. What is the interaction length of the tilted-wavefront pump on the LN? Can the spot of the
pump on the LN be imaged by the CCD camera?

**Response1:** Thank the reviewer for this question. It is difficult to directly measure the
interaction length, so we give an estimation according to the experimental results. According
to our experiment, the interaction length of the tilted-wavefront pump on the LN is about 0.89
196 mm - the distance that THz waves propagate in about 10 ps. Here, we also considered that the
197 pump laser does cause electro-optic effect and can be imaged by the CCD camera. Therefore,
in the experiment, we did not analyze the signal before the pump laser leaves the waveguide
for the validity of our results. Specifically, in the spatial choice, only the data after pump spot
was used; in the temporal choice, only the data after pump laser leaving the waveguide is
considered. Thus, the potential influence of the pump laser on the experimental signal is
avoided.

We added this estimated interaction length of 0.89 mm to the main text (Page 4 Lines
 127~128).

2. In Eq. (S3-2), it is strange to use an exponential decay function to describe the effective
 intensity of the pump laser. Shouldn't the pump spot be similar to Gaussian like function
 (rise-peak-decrease) rather than monotonically decay function? What value of Gamma was
 used for the fitting results?

**Response2:** Thank the reviewer for this comment. According to the experimental setup, the
 effective decay of the pump laser is indeed more similar to a Gaussian-like function rather
 than an exponential decay function. In the last version, we used an exponential decay
 function to simplify the model and the calculation. Here, we agree with the reviewer.
 However, since the indefinite integral of Gaussian function does not have an analytic form,
 we consider a "piecewise exponential function" (Eq. (S2) in the Supplementary Materials) of

$$dE_0 = I_{G0} \exp(-\Gamma|x - d|) dx - \alpha_0 E_0 dx,$$

$$dE_1 = I_{G1} \exp(-\Gamma|x - d|) dx - \alpha_1 E_1 dx,$$

to describe the variation of the pump intensity, and the fitting is also reperformed. In the new
 fitting, the best fitting parameters are $I_{G0}=4.7023$ kV/mm², $I_{G1}=4.3944$ kV/mm², $\Gamma=1.2787$
 220 mm⁻¹, and $d=0.65$ mm. The new fitting results are shown as Fig. S1 in Supplementary
 Materials (also shown below). We hope these new results could be more convincing.

Fig. S1. Plot of the electric field of THz waves as a function of position x . The solid curves
 show calculated results and the circles are from experimental results. The error bar is
 evaluated from the fitting deviation and the experimental noise. E_0 , E_1 , and E_d are the electric
 field amplitudes of THz waves with frequency ν_0 , ν_1 , and ν_d , respectively.

3. In Eq. (S3-3), ISRS should also be considered. E_d should not be only from the amplitudes
 of the two waveguide frequency modes.

**Response3:** According to above discussion (Response0, Lines 143~187), E_d is mainly from
 the DFG, and the contribution of ISRS is ignorable. In addition, this effect was excluded by
 just analyzing the experimental data after the pump laser left the waveguide (Response 1,
 Lines 192~201).

 Therefore, we ignored the contribution of ISRS in Eq. (S3-3) [Revised version: Eq. (S5) in
 the Supplementary Materials)]. The fitting results in Fig. S1 also verified the validity of this
 argument.

 4. What exact function was the applied driving THz field $E_T(t)$ taken for the calculation
 through Eq. (4)-Eq.(8) to obtain the theoretical $\text{Chi}(2)_{pp} = 1.7 \times 10^{-5}$ m/V? Does the
 calculated $\text{Chi}(2)$ depend on different amplitudes of the applied driving THz field?

 **Response4:** Thanks for the question. The second-order nonlinearity ($\chi^{(2)}$) is an intrinsic
 property of a material in an optical system, which is independent of the external field, as in
 Ref. 1 [R. W. Boyd, Nonlinear Optics. Elsevier (2003)]. In solving Eq. (4) to Eq. (8), we also
 referred to the methods used in nonlinear optics, where $E_T(t)$ makes no difference. For the
 convenience of the reviewer, we give the solving steps here:

 First, we start from Eq. (4) and assume $x = x_1 + x_2$, with x_1 and x_2 represents the linear and
 nonlinear response of the external driving field $E_T(t)$, and thus

$$-\omega^2 x_1 - j\gamma\omega x_1 = \left(\frac{q}{m}\right) E_T(t),$$

$$-\omega^2 x_2 - j\gamma\omega x_2 = -a(x_1 + x_2)^2 \approx -ax_1^2.$$

where the approximation $(x_1 + x_2)^2 = x_1^2 + 2x_1x_2 + x_2^2 \approx x_1^2$ is valid because generally
 $x_1 > x_2$ [Ref. 1 (R. W. Boyd, Nonlinear Optics. Elsevier (2003))], and then we can solve x_1
 and x_2 as

$$x_1(\omega_1) = \frac{1}{D(\omega)} \left(\frac{q}{m}\right) \tilde{E}_T(\omega_1),$$

$$x_2(\omega_1 - \omega_2) = -\frac{a}{D(\omega_1 - \omega_2)} x_1(\omega_1) x_1(-\omega_2)$$

$$= -\frac{a}{D(\omega_1 - \omega_2) D(\omega_1) D(-\omega_2)} \left(\frac{q}{m}\right)^2 \tilde{E}_T(\omega_1) \tilde{E}_T^*(\omega_2).$$

Here, we write $E_T(t)$ in frequency domain as $\tilde{E}_T(\omega)$ for calculation. The linear susceptibility
 and the second-order susceptibility are defined through the relations

$$P^{(1)}(\omega_1) = \epsilon_0 \chi^{(1)}(\omega_1) E(\omega_1)$$

$$P^{(2)}(\omega_1 - \omega_2) = \epsilon_0 \chi^{(2)}(\omega_1 - \omega_2; \omega_1, \omega_2) E(\omega_1) E^*(\omega_2)$$

Since the contributions to the polarizations are

$$P^{(1)}(\omega_1) = -Nex_1(\omega_1)$$

$$P^{(2)}(\omega_1 - \omega_2) = -Nex_2(\omega_1 - \omega_2)$$

Therefore, the linear susceptibility and the second-order susceptibility are

$$\chi^{(1)}(\omega_1) = \frac{Nq^2}{\epsilon_0 m} \frac{1}{D(\omega_1)}$$

$$\chi^{(2)}(\omega_1 - \omega_2; \omega_1, \omega_2) = \frac{Nq^3}{\epsilon_0 m^2} \frac{a}{D(\omega_1 - \omega_2)D(\omega_1)D(-\omega_2)}$$

Here, we can see that both the second-order susceptibility and the linear susceptibility are
 independent of the external field. Therefore, in the calculation, we did not need to consider
 the specific value of the driving field $E_T(t)$.

5. Pump-power dependent measurement should be helpful to clarify the generation
 mechanism of the difference frequency. The relation between the intensity of the difference
 frequency and the pump power should be different for generation through ISRS and
 generation purely from two different waveguide modes. The theory proposed in this
 manuscript can also be verified by pump-power dependent measurement.

**Response5:** Thank the reviewer for the advice. Admittedly, the power dependence of ISRS
 generation and DFG generation is different. While the pump-power dependent measurement
 is helpful, as discussed above in Response0 (Lines 143~187), it is not necessary and not the
 main focus of this paper, since we have a sound and self-consistent discussion to show that
 ISRS generation has less possibility to exert influence on the difference frequency signal.
 Nevertheless, we appreciate the reviewer's suggestion, and the pump-power dependent
 measurement, along with pump-frequency dependent measurement, pump-polarization
 dependent measurement, and other relevant measurements will be performed in our future
 works to explore other fundamental phenomena and potential applications mediated by
 phonon-polariton nonlinearity.

We thank the reviewer again for the critical review, and sincerely hope the reviewer can be
 satisfied with our response.

Reviewer #3 (Remarks to the Author):

In this work Lu et al. demonstrate the difference frequency generation (DFG) of two THz
 waves generated with a femtosecond pump laser through the impulsive stimulated Raman
 scattering process in a Lithium Niobate (LN) waveguide. They estimate a giant 2nd order
 nonlinear susceptibility, χ_2 , in the order of $10^{(-6)}\text{m/V}$, which is attributed to phonon-
 polaritons in the LN crystal. While there are previous works on THz wave generation (Ref 42)
 and DFG of near infrared input signals (Sasaki & Yuri, Appl. Phys. Lett. 81, 3323, 2002) in
 LN, as far as I know this is the first observation of DFG where both the input and output
 signals are in the THz range. I found the experiment very interesting, the evidence for the
 DFG signal convincing, and the time-resolved imaging of the THz waves in the video
 beautiful. The giant value of χ_2 makes LN a promising material for frequency conversion
 of THz signals, which is useful for many applications including optical control of spin qubits

in semiconductors. However, I think there are many important issues that need to be
addressed before the paper is suitable for publication in Nature Communication.

**Response0:** We thank the reviewer for all the positive comments. We carefully read and
studied the previous work recommended by this reviewer (Sasaki & Yuri, Appl. Phys. Lett.
81, 3323, 2002). We found it very helpful to clarify the novelties and advances of our work in
simplifying the experimental setups, optimizing phase-match conditions, demonstrating the
dynamic process of the frequency-mixing process, and enhancing the THz nonlinearity by
phonon polaritons, so it has been included in our reference list (Ref. 14 in the revised
version). Here, the reviewer's comments also reminded us to consider the potential
applications of such a giant nonlinearity in applications such as optical control of spin qubits
in semiconductors. Since the spin qubits tunability of the exchange coupling rate ranges from
many GHz to THz (M. D. Reed, et al., Phys. Rev. Lett. 116, 110402 (2016)), it is probable to
use such a giant nonlinearity to control the spin qubits and further to optimize the quantum
computations. Accordingly, we also included the relevant discussions in the revised
manuscript (Page 8 Lines 272~273). Besides, we also tried our best to address the important
issues that the reviewer pointed out, and we hope the revised version is now suitable for
publication in Nature Communication.

1. The estimation of the experimental χ^2 can be presented in a more transparent way. In the
Methods χ^2 is stated to be obtained from "solving Eq. S3-2 and S3-3." Given the
importance of χ^2 in this work, I think the authors should express clearly how χ^2 depends
on the experimental and fitting parameters. This does not seem hard to do, since Eq S3-2 has
an analytical solution. A more detailed description of how the experimental χ^2 is estimated
should also be provided in the main text.

**Response1:** Thank the reviewer for the advice. The experimental $\chi^{(2)}$ is a critical result to
clarify the giant second-order nonlinearity. Here, we totally agree with the reviewer, and
more details about how to evaluate the results have been provided in the revised manuscript.
Moreover, some necessary contents were added in the revised manuscript both in the main
text (Page 5 Lines 174~183) and the Supplementary Materials (Page 2 Lines 43~71).

According to Reviewer 2 (Comment 2, Line 205), we changed the decay model of the pump
laser, in the revised version, the specific process to solve $\chi^{(2)}$ is provided. Specifically,
according to our assumption, we can obtain Eq. (S2) in the Supplementary Materials, where
the function $\exp(-\Gamma|x-d|)$ with parameters Γ and d represents the power change of the
pump laser pulses. Here, we assume $0 < d < x_1$ according to the experimental results. For
the convenience of the reviewer, we copy the relative contents here as follows.

Original Text (Page 2 Lines 43~71 in the Supplementary Materials)

"It is assumed that the THz electric fields for the three frequencies at the position x are
$E_0(x)$, $E_1(x)$, and $E_d(x)$, respectively. Then consider the change in the E -field from position

x to $x + dx$. The change in the field E_0 and E_1 mainly comes from the generation of
 femtosecond laser and the material absorption, which indicates that

$$\begin{aligned} dE_0 &= I_{G0} \exp(-\Gamma|x - d|) dx - \alpha_0 E_0 dx, \\ dE_1 &= I_{G1} \exp(-\Gamma|x - d|) dx - \alpha_1 E_1 dx, \end{aligned} \quad (S2)$$

where I_{G0} and I_{G1} depend on the effective intensity of the laser pulse due to the properties of
 impulsive stimulated Raman scattering. The function $\exp(-\Gamma|x - d|)$ with parameters Γ and
 344 d represents the power change of the pump laser pulses. Here, we assume $0 < d < x_1$
 according to the experimental results.

According to the initial value of $E_0(x = 0) = E_1(x = 0) = 0$, here $E_0(x)$ and $E_1(x)$ can
 be analytically solved as

$$\begin{aligned} E_0 &= e^{-\alpha_0 x} \int_0^x I_{G0} e^{\alpha_0 \xi - \Gamma|\xi - d|} d\xi, \\ E_1 &= e^{-\alpha_1 x} \int_0^x I_{G1} e^{\alpha_1 \xi - \Gamma|\xi - d|} d\xi, \end{aligned} \quad (S3)$$

By fitting all the spectra of THz waves as in Fig. 4b, the electric field of THz waves at
 different frequencies and different positions can be obtained, as shown in Table S1. Using a
 similar method as what we used in Fig. 5, we can get the boundary condition to evaluate the
 fitting parameter in Eq. (S3).

Table. S1 The electric fields of THz waves at different frequencies and different positions.

	E_0 (kV/mm)	E_1 (kV/mm)	E_d (kV/mm)
x_1	3.29	1.71	0.935
x_2	3.659	1.26	0.453
x_3	3.67	0.61	---
x_4	2.74	0.24	---

353 We solve the values of the parameters and obtain that

$$\begin{aligned} I_{G0} &= 4.70 \text{ kV/mm}^2; \\ I_{G1} &= 4.39 \text{ kV/mm}^2; \\ \Gamma &= 1.28 \text{ mm}^{-1}; \\ d &= 0.650 \text{ mm}; \end{aligned} \quad (S4)$$

The change in E_d signal is mainly due to the DFG, material absorption, and waveguide
 mode attenuation. Thus

$$dE_d = E_{\text{DFG}} - \alpha_d E_d dx - \Gamma_M E_d dx,$$

$$E_{\text{DFG}} = p\chi^{(2)} \frac{2\pi\nu_d E_0 E_1}{cn_{\text{eff}}} dx, \quad (\text{S5})$$

where E_{DFG} stands for the DFG of the field E_d signal by E_0 and E_1 . The initial and boundary
 conditions are considered to be

$$E_d(x = 0) = 0;$$

$$E_d(x = 1.18) = 0.935 \text{ kV/mm}. \quad (\text{S6})$$

The units used above are millimeter (mm), kilovolt (kV), and picosecond (ps). Substituting
 E_0 and E_1 in Eq. (S5) with Eq. (S3) and the parameters in Eq. (S4), we can solve Eq. (S5).
 The value of the second-order nonlinear susceptibility can be obtained by using the boundary
 condition Eq. (S6), and the result is $\chi^{(2)} = 1.58 \text{ mm/kV} = 1.58 \times 10^{-6} \text{ m/V}$.

Substituting Eq. (S5) with this result, we get the solution of $E_d(x)$. The corresponding
 electric fields dependent on position x are shown in Fig. S1. Three solid curves show the
 calculated results of $E_0(x)$, $E_1(x)$, and $E_d(x)$, respectively, which agree well with the
 experimental results marked with symbols.”

 2. There are three important fitting parameters used in the estimation of the experimental chi2:
 The effective intensities I_{G0} , I_{G1} , and the decay constant Γ , introduced in Eq. S3-2 for
 explaining why the lower-frequency THz wave (ν_0) at first increases with the distance of
 propagation up to $x_3=2.39\text{nm}$. I could not find the fitted values of these parameters in the
 paper. These values should be stated along with a justification for why they are reasonable
 and consistent with the underlying physics of the impulsive stimulated Raman scattering
 process. Moreover, what is the value of the waveguide mode attenuation M in Eq. S3-2? Is it
 another fitting parameter?

 **Response2:** In the last version, we skipped the many fitting details. Here, we agree with the
 reviewer that the fitting parameters of I_{G0} , I_{G1} and Γ are very important to estimate the
 experimental $\chi^{(2)}$, and thus we added more details about the fitting parameters in the revised
 manuscript. In our analysis, the fitting calculation is performed as follows:

 (1) In the Eqs. (S3-2) and (S3-3) [Revised version: Eqs. (S2) and (S3) in the
 Supplementary Materials], we have three fitting parameters, I_{G0} , I_{G1} , and Γ . We take
 the field amplitudes of E_0 , E_1 , and E_d at $x_1 = 1.18 \text{ mm}$ as the boundary values of the
 fitting. The parameter values and the reasonability of the fitting are verified by the
 comparison between the solution of these two equations and the other experimental
 data, the field amplitudes of E_0 , E_1 , and E_d at x_2 , x_3 , and x_4 .

(2) The mode attenuation M , which has been changed to Γ_M according to Comments 9
(Line 582), is not a fitting parameter. The attenuation of waveguide mode Γ_M was
evaluated by the phase change during propagating back and forth in the orthogonal
direction in the waveguide. As seen in our previous work [Ref. 44 (C. Yang, et al.,
Opt. Express 18, 26351 (2010))], the waveguide mode requires this phase shift to be
an integer multiple of 2π , while it would be an arbitrary value when not in any modes
of the waveguide. This phase could be used to evaluate the mode attenuation because
an arbitrary phase enables us to perform a vector superposition to calculate the
practical propagation. Here the phase is calculated through the mode equation [Ref.
44 (C. Yang, et al., Opt. Express 18, 26351 (2010))] of the waveguide, which is
shown in Eq. (S1). After substituting the known values, the phase change is calculated
to be 0.92π . As we explained in the manuscript, the effective loss by mode mismatch
399 per millimeter is $\Gamma_M E_d = 20.43 E_d$, where E_d is the electric field of THz waves with
400 frequency ν_d .

During the fitting, the best results are shown in Fig. S1 in the Supplementary Materials, and
the fitting parameters are explained above and also in the revised manuscript. In the new
fitting, the best fitting parameters are $I_{G0}=4.7023$ kV/mm², $I_{G1}=4.3944$ kV/mm², $\Gamma=1.2787$
405 mm⁻¹, $d=0.65$ mm. The new fitting results are shown in Fig. S1 in the revised version. We
hope these new results are convincing enough.

3. The authors claimed that the value of chi2 in their work is many orders of magnitude larger
than those obtained for visible light and microwave but did not cite any references. Their
claim could be strengthened by a table of comparison between their chi2 and those reported
for other materials in the literature. I am not sure why the authors compare their result with
only visible and microwave values. A comparison with chi2 of other materials in the THz and
infrared ranges is more relevant.

**Response3:** Thank the reviewer for the advice. In the last version, we just compare our
results with the one at microwave and visible light in the LN crystals to show the contribution
of phonon polaritons at THz frequency. The reviewer's suggestion reminded us of the
necessity to make a comparison between our result and the previous results in other materials
and similar frequencies. Therefore, a new table is added in the revised version to make it
more specific and clearer. (see Pages 5~6 Lines 191~196 in the main text and Table S2 in the
Supplementary Materials.)

From Table S2 in the Supplementary Materials, it can be seen that the phonon-polariton
enhanced LiNbO₃ crystal has the largest nonlinear susceptibility, several orders of magnitude
larger than that in traditional semiconductor crystals, organic crystals, and metasurfaces.

4. The significance of the result, especially its relevance for THz applications, should be
given more discussion. It is stated in the paper that the giant chi2 of LN can be useful for
429 THz technology. Unlike in this experiment where both the input and output THz signals are

430 generated inside the crystal, in most frequency conversion applications the input signals are
431 generated from an external source, for example a quantum cascade laser. Is it possible to
432 design a LN waveguide for DFG and SFG (sum-frequency generation) of external inputs in
the THz range? In this experiment strong loss is observed for the output DFG signal, which
may limit its uses. Is there a way to reduce the loss?

**Response4:** Thanks again for the helpful and constructive advice. Indeed, the giant second-
order nonlinearity at THz frequency is important for a variety of THz applications. Although
in our experiment, both the input and output THz signals are generated inside the crystal, this
giant-nonlinearity mediated by phonon polaritons still works when the THz waves are
launched from external sources, just like QCL (quantum cascade lasers). Furthermore, using
better monochromatic THz sources would obtain a higher efficiency because the interaction
distance could be much longer. This may also provide possibilities to nonlinear technologies
at THz frequencies. However, there are also some questions to solve when using external
444 THz sources. In our experimental setups, the phase-match condition is satisfied because both
the two waveguide modes match with the pump laser, while the phase-match condition would
become a very important question that needs to be addressed when external THz sources
were applied to achieve DFG or SFG. The relevant discussions have also been added to our
revised manuscript (Page 8 Line 259~265 in the main text).

For reducing the loss, we could discuss it from the following aspects:

(1) Materials absorption: material absorption mainly comes from resonance absorption
and defect absorption. The resonance absorption is caused by the heat-induced
spontaneous resonance of the ions in the crystal, and it can be largely suppressed by
lower down the temperature (the temperature-dependent absorption can be seen in Ref.
45 [X. Wu, et al., *Opt. Express* 23, 29729 (2015)]). In addition, the defects in the
crystal can also cause absorption, which would be gradually reduced as the crystal
growing technology develops.

(2) Mode attenuation: the mode attenuation is easy to control by mode design. In our
work, we choose the DFG signal mismatch the waveguide modes, is to ensure the
signal comes from DFG, rather than ISRS of the pump laser. However, when an
external source was applied, it is possible to design a waveguide that could support
both the input and output signals in certain of its modes, which could eliminate or
remarkably reduce the mode attenuation of the waveguide mode.

(3) Furthermore, to choose a very thin subwavelength waveguide could also reduce the
absorption, because where more energy of the waveguide mode can remain outside of
the waveguide (air) as an evanescent wave, see more details in our previous work in
Ref. 44 [C. Yang, et al., *Opt. Express* 18, 26351 (2010)].

5. For completeness previous works on THz DFG in LN, for example Sasaki & Yuri, *Appl.*
*Phys. Lett.* 81, 3323, 2002, should be discussed. It would be great for researchers in the field
if the authors can make clear what the advances are in their work compared with previous
works on THz DFG.

**Response5:** Thank the reviewer here. The work “Sasaki & Yuri, Appl. Phys. Lett. 81, 3323
(2002)” is definitely important. We agree with the reviewer that we must discuss that work to
make our work clearer and show our advances by comparison. The paper in 2002 reported a
477 THz wave DFG in “slant-stripe-type PPLN” between two signal waves, which was generated
in two different kinds of phase-match conditions by a nanosecond Nd:YAG laser. That work
is very important and the author was aimed to solve the THz absorption during its generation,
whose results were well demonstrated. In the revised version, we have added necessary
discussions in Page 2 Lines 39~41, 71~73, and included this paper in our reference list (Ref.
14 in the revised version).

Compared with their work, our work achieved a giant second-order nonlinear process
mediated by phonon-polaritons. All the theoretical analyses are dependent on the nonlinear
modified Huang equations (Eqs. (1-1) and (1-2)) rather than the enharmonic Lorentz
oscillator equation. Besides, we presented a fine time-resolved imaging of the DFG process,
which first showed a dynamic view of the frequency-mixing process. In addition, we
integrate the generation of THz waves and the DFG process in a single crystal, which
remarkably simplifies the experimental setup and provides possibilities for future on-chip
491 THz technologies.

Therefore, the giant nonlinearity we reported would make it much easier to achieve
supercontinuum spectra or frequency combs at THz frequency, so as to construct an on-chip
integrated platform which may benefit numerous physical, chemical, and biological systems
based on THz technology, including optical control of spin qubits in semiconductors (as the
reviewer reminds us before). Our work may also stimulate a large number of new studies
based on phonon polariton-induced nonlinearity at THz frequency, such as to construct
efficient optomechanical platforms or perform various modulations to the properties of ionic
crystals, including nonlinear susceptibilities, phase transitions, and domain structures.

I also have a few minor concerns and questions below:

6. In the analysis the authors used a formula for monochromatic inputs for E_{DFG} in Eq. S3-
3. However, it is shown in Fig. 4b that the input THz waves are pulses with a significant
frequency spread of about 0.1 THz. In this case E_{DFG} is given by an integral over the
different frequency components of the inputs. The effect of this spectral width may be
significant, as it is comparable to the central frequency of the lower-frequency THz wave.
Can the authors comment on why the monochromatic approximation is valid for the
estimation of the experimental χ^2 ?

**Response6:** Thanks for this comment. The reviewer concerns about the validity of the
monochromatic approximation. We also considered the possible influence of the pulsed THz
waves (frequency spreading) on our results. Indeed, the DFG signal should be calculated by
an integral over the different frequency components of the inputs. Specifically, the electric
fields of the DFG signal should be

$$\begin{aligned}
E_{\text{DFG}}(\omega) &= \int_{\omega_{\text{d,min}}}^{\omega_{\text{d,max}}} E_d(\omega_d) d\omega_d \\
&= \int_{\omega_{\text{d,min}}}^{\omega_{\text{d,max}}} \int_{\omega_{0,\text{min}}}^{\omega_{0,\text{max}}} \int_{\omega_{1,\text{min}}}^{\omega_{1,\text{max}}} \chi^{(2)}(\omega_d; \omega_0, -\omega_1) \frac{\omega_d}{cn_{\text{eff}}} E_0(\omega_0) E_1^*(\omega_1) \\
&\quad \times \exp[i\Delta k(\omega_d, \omega_0, \omega_1)x] d\omega_0 d\omega_1 d\omega_d
\end{aligned}$$

Where Δk represents the phase-match condition. In the calculation based on the peak value,
the phase-match condition is nearly perfect satisfied, because the input THz waves are
generated through the velocity match between the pump laser and the waveguide modes. It is
worth noting that although the phase-match condition is well satisfied at the peak value, it
should be reconsidered when considering the different frequency components around it. For
simplicity, we only consider the peak value of difference frequency here, and it could be
written as

$$\begin{aligned}
E_{\text{DFG}}(\omega_d) &= \int_{\delta\omega_{\text{min}}}^{\delta\omega_{\text{max}}} \chi^{(2)}(\omega_d; \omega_0 + \delta\omega, -\omega_1 - \delta\omega) \frac{\omega_d}{cn_{\text{eff}}} E_0(\omega_0 + \delta\omega) E_1^*(\omega_1 \\
&\quad + \delta\omega) \exp[i\Delta k(\delta\omega)x] d\delta\omega
\end{aligned}$$

In this equation, we use an integral of $\delta\omega$ to calculate all the frequency components that can
result in ω_d by DFG over all the input wave packets. Here, the phase-match parameter is a
function of $\delta\omega$, which is easy to understand from Fig. 2. Clearly, the perfect phase-match
condition here would be broken as the absolute value of the deviation frequency $|\delta\omega|$
increases.

Specifically, the perfect phase-match condition requires the wavevector $\Delta k = k_1(\omega_1) -$
$k_0(\omega_0) - k_d(\omega_d) = 0$. While if $\delta\omega \neq 0$, the wavevector
 $\Delta k' = k_1(\omega_1 + \delta\omega) - k_0(\omega_0 - \delta\omega) - k_d(\omega_d)$

$$\approx \left[k_1(\omega_1) + \delta\omega \frac{dk_1}{d\omega_1} \right] - \left[k_0(\omega_0) - \delta\omega \frac{dk_0}{d\omega_0} \right] - k_d(\omega_d) = \delta\omega \left(\frac{dk_1}{d\omega_1} + \frac{dk_0}{d\omega_0} \right),$$

This phase mismatch would largely decrease the DFG efficiency as the absolute value of the
deviation frequency $|\delta\omega|$ increases, since both $\frac{dk_1}{d\omega_1}$ and $\frac{dk_0}{d\omega_0}$ is a large positive number (see
Fig. 2(c) in the main text). Furthermore, when deviation frequency increases, the power of
input THz wave in the wave packet also largely decreases. Therefore, we think the
calculations based on the peak value of the THz field is reasonable to estimate the second-
order susceptibility.

Moreover, in our calculation, we considered the frequency spread in the time domain. As a
result of the frequency spread, both the input THz waves and the output THz waves are not
infinite plane waves in the time domain, but behave as pulses. Therefore, a spatial “walk-off
effect” would occur, which makes this calculation more complicated. In our model, the
contribution of the walk-off effect is estimated by setting a parameter $p(x)$, and this
parameter makes a good agreement with such “frequency spreading” phenomena.

Besides, both the fitting results in Fig. S1 and the solution to nonlinear modified Huang
equations also well verified the validity of our calculation.

7. In the third paragraph of the Introduction it is stated that the strong nonlinearity in Refs 26-
34 is due to “high mobility of the electrons.” Mobility is defined for charge carriers. In some
of the cited works, for example Refs 26 and 28, strong nonlinearity is observed for electrons
in bound states where mobility is not a defined concept. The strong nonlinearity in these
systems comes from the large extent of the wavefunction resulting in a huge dipole coupling
with fields.

**Response7:** Thank the reviewer for reminding us. After carefully check the works that the
reviewer mentioned, we agree with the reviewer that those works, the strong nonlinearities
are not caused by “high mobility”. Specifically, in M. A. W. van Loon, et al., Nat. Photon. 12,
179 (2018), the authors reported a giant multiphoton absorption through tuning the dipole
moment through a THz free-electron laser, while in V. Walther, et al., Nat. Commun. 9, 1309
(2018), the giant nonlinearities come from Rydberg excitons in the semiconductors.
Therefore, a special note was added in the revised version (Page 2 Line 60).

8. What are the beam radii of the pump and probe femtosecond lasers? These are necessary
for estimating the intensities and the electric fields of the pump and the probe.

**Response8:** Thank the reviewer for this question. In our experimental setup, the beam radius
of the probe laser is rather large (about 8 mm) to imaging the whole sample, and the beam
radius of the pump laser is about 0.5 mm. These two values are also added in the revised
manuscript (Page 8 Lines 281 & 282), and we hope it could help to make our experiment
clearer.

9. The notation M is used for both the magnification of the imaging system and the
waveguide mode attenuation.

**Response9:** Thanks for the critical reading. It is our mistake to use notation M twice. In the
revised version, the notation M is only used for the magnification of the imaging system. The
waveguide mode attenuation is now indicated by a new notation Γ_M . We believe this change
will make it clearer.

We thank all reviewers again for the critical review and constructive comments, which
helped us to improve the manuscript greatly. We hope the reviewers can now accept our
revised manuscript for publication in Nature Communications.

REVIEWER COMMENTS

Reviewer #1 (Remarks to the Author):

The authors thoroughly addressed my concerns and made significant improvements to the manuscript. I also think that that the remarks raised by the other reviewers have been adequately addressed. At this stage, I recommend publication of this work in its present form.

The authors properly addressed some of the issues I raised, but did not satisfactorily address the core issue about if the non-waveguide mode is dominated by the $\chi^{(2)}$ process of the two waveguide modes at THz frequency. The authors thought it is not necessary and not the main focus of this paper to do pump-power dependent measurement although they claimed it will be done for future work. Conversely, I argue that pump-power dependence measurement (even just preliminary data of two different powers) is important to address this issue without ambiguity and should be done to strongly support the whole content in this manuscript.

I plot the dispersion relation curve of the velocity match of the laser pulses (in black dotted line) on Fig. 2c as follows. The signal, not supported by any waveguide mode, also lies on the line of the velocity match of the laser pulses. Since it lies on the matching line of the pump laser, it is possible that the signal of this mode can be continually contributed from the pump laser pulses. And the effect of the pump laser pulses could last until $x = 2.39$ mm which is within the analysis of the non-waveguide mode signals. Fig. S1 reveals that E_0 increases until $x=2.39$ mm, which implies that the contribution from the pump laser could last until $x=2.39$ mm. But E_d only shows significant signals at $x_1=1.18$ and $x_2=1.82$ mm. According to the fitting parameters in Eq. (S4) that $\Gamma = 1.28 \text{ mm}^{-1}$ and $d=0.65$ mm, the generation amplitude drops to $\sim 0.22I_G$ at $x=1.82$ mm compared with the peak value I_G at $x=0.65$ mm. It seems that within 2 mm, the effect of pump laser still remains until E_d disappears.

In addition, if $\chi^{(2)}$ at THz range is so giant, why didn't second harmonic generation of ν_0 significantly appear in this results? As mentioned by the authors, the DFG process $\chi^{(2)}$ ($\nu_d = \nu_1 - \nu_0$) encounters the walk off effect because of different group velocities.

However, the SHG $\chi^{(2)}$ ($2\nu_0 = \nu_0 + \nu_0$) should not suffer from the walk-off effect. The square of E_0 is also larger than the product of E_0 and E_1 . As a result, signals from SHG is expected to be larger than that from DFG if $\chi^{(2)}$ of this two processes are on the same orders. But in Fig. 4b, it is puzzling that the signal at $2\nu_0 = 0.7$ THz is much weaker than that of $\nu_d = 0.76$ THz.

Appl. Phys. Lett. 99, 071102 (2011) demonstrated high power tunable multicycle THz pulses also in LN. The pump pulse energy (6 mJ and 35 mJ) is order of magnitude higher than the pulse energy (0.45 mJ) in this manuscript. It should have stronger THz field that could generate larger $\chi^{(2)}$ signals if $\chi^{(2)}$ at THz range is so giant as the authors claimed. Their results at two different pump power are as follows.

[Redacted]

With high pump energy of 35 mJ, it may have SHG component. However, with 6 mJ, the SHG frequency component is not obvious since the power dependence of $\chi^{(2)}$ signal is nonlinear. In contrast, it is very surprising that the authors in this manuscript observed “nonlinear signals” with pump energy as low as 0.45 mJ. It seems that the $\chi^{(2)}$ for SHG at THz range from the results of APL 99, 071102 (2011) is much lower than the $\chi^{(2)}$ for DFG at THz range the authors claimed in this manuscript.

In summary, the effect of pump laser may last until E_d disappears. Second, it is puzzling that DFG overwhelms SHG in the data of this manuscript. Third, the authors observed “nonlinear signal” with pump pulse energy order of magnitude low compared with the results of APL 99, 071102 (2011). I sincerely hope the authors are convinced that pump-power dependence measurement (even with only two different powers) is necessary to strongly support the argument that contribution to E_d is completely from the nonlinear $\chi^{(2)}$ process of DFG in this manuscript.

Reviewer #3 (Remarks to the Author):

The authors have addressed all my concerns in the revisions. While the experimental nonlinear susceptibility was estimated based on four fitting parameters and a few assumptions, such as the "piecewise exponential function" for the pump laser and the monochromatic approximation, I believe the precise detail of the fitting procedure has no significant effect on the order of magnitude of the reported nonlinear susceptibility. Therefore, the authors' claim of a giant nonlinear susceptibility many orders of magnitude larger than previously reported values is valid. This is an interesting experimental result that deserves publication in Nature Communications.

Response to the Reviewers:

Reviewer #1:

The authors thoroughly addressed my concerns and made significant improvements to the manuscript. I also think that the remarks raised by the other reviewers have been adequately addressed. At this stage, I recommend publication of this work in its present form.

Response: We wish to express our sincere gratitude and appreciation to the reviewer for his/her time and the positive recommendation for our manuscript. The previous comments and suggestions helped us greatly in improving the manuscript.

Reviewer #2:

1. The authors properly addressed some of the issues I raised, but did not satisfactorily address the core issue about if the non-waveguide mode is dominated by the $\chi^{(2)}$ process of the two waveguide modes at THz frequency. The authors thought it is not necessary and not the main focus of this paper to do pump-power dependent measurement although they claimed it will be done for future work. Conversely, I argue that pump-power dependence measurement (even just preliminary data of two different powers) is important to address this issue without ambiguity and should be done to strongly support the whole content in this manuscript.

Response1: We are glad the other concerns from the reviewer are well addressed. The reviewer here mainly cares about the influence of laser generation on the $\chi^{(2)}$ process, and strongly recommended us to perform a pump-power dependent experiment. Here, we present two additional experimental results and a theoretical analysis to prove the validity of our claim.

The dispersion curves and the spectral information of the two *new* results are provided in Fig. R1 and Fig. R2, respectively. The original result in the main text is also provided in Fig. R3 for reference and direct comparison.

Fig. R1 (Marked as Experiment 1) (a) Experimental dispersion curve under a low pump power (about 81% smaller than the main-text value). (b) The representative spectrum in which the waveguide modes do not walk off. The DFG signal can be hardly seen.

Fig. R2 (Marked as Experiment 2) (a) Experimental dispersion curve under a high pump power (about 43% larger than the main-text value). The DFG and sum frequency generation (SFG) signals can be clearly seen. (b) The spectrum where the amplitudes of the DFG signal are large ($x = 1.16$ mm).

Fig. R3 (Original main-text experiment) (a) (Fig. 2c in the main text) Experimental dispersion curve in the main-text. The DFG signal is clearly visible and stronger as compared to Fig. R1.

(b) (Fig. 5a in the main text) The spectrum in which the amplitudes of the DFG signal are large ($x = 1.18$ mm).

In the above two new results, the Experiment 1 is performed under a low-pump power (about 81% smaller than the main-text value), where the field amplitudes of the matched zero-order ($\nu'_0 \approx \nu_0 = 0.34$ THz) and first-order ($\nu'_1 \approx \nu_1 = 1.1$ THz) waveguide modes have similar frequencies as the results in the main text. However, the field amplitudes of the waveguide modes are much smaller than the values in the main text. In this case, the DFG signal is very weak and hardly to be seen in the spectra, as shown in Fig. R1(b). This result indicates that the DFG signal does depend on the two waveguide modes nonlinearly, and the nonlinear frequency-mixing process dominates the generation of E_d . Only in the dispersion curve can we see the weak DFG signal, as Fig. R1(a) indicated, and a very small color-bar limit is used here in order to identify and distinguish the DFG signal. While the noise appears in the low-wavevector regime, thus the DFG signal is hardly seen in Fig. R1(b). Nevertheless, the noise causes little influence when the THz field is strong enough, as shown in Fig. R2(a) and Fig. R3(a).

As Fig. R2 shows, the Experiment 2 is performed under a high-pump power (about 43% larger than the main-text value), where the field amplitudes of the matched waveguide modes are larger than the values in the main text. In order to further verify the difference-frequency relation, we slightly change the wavefront tilt angle α . Then the zero-order ($\nu''_0 = 0.42$ THz $>$ $\nu_0 = 0.34$ THz) and first-order ($\nu''_1 = 1.31$ THz $>$ $\nu_1 = 1.1$ THz) have larger frequencies than the values in the main-text experiment, the DFG signal here also satisfies $\nu''_d = \nu''_1 - \nu''_0$. Using the same method (gaussian-function fitting) in the main text, the fitting amplitudes of the signals at $x = 1.16$ mm and $x = 1.31$ mm are calculated to be:

$$\begin{aligned} E(\nu''_0, x = 1.16) &= 6.216 \text{ kV/mm}; E(\nu''_0, x = 1.31) = 6.382 \text{ kV/mm}; \\ E(\nu''_1, x = 1.16) &= 2.926 \text{ kV/mm}; E(\nu''_1, x = 1.31) = 2.980 \text{ kV/mm}; \\ E(\nu''_d, x = 1.16) &= 2.129 \text{ kV/mm}; E(\nu''_d, x = 1.31) = 1.682 \text{ kV/mm}. \end{aligned}$$

Considering the material absorption for THz waves at the frequencies ν''_0 , ν''_1 , and ν''_d are $\alpha''_0 = 0.35 \text{ mm}^{-1}$, $\alpha''_1 = 2.1 \text{ mm}^{-1}$, and $\alpha''_d = 1.1 \text{ mm}^{-1}$ respectively, we constructed a similar model as we used in the main text. After calculation, the nonlinear susceptibility shows a value of $\chi^{(2)} > 1.352 \text{ mm/kV}$. Correspondingly, the theoretical value in these frequencies given by Eq. (4) in the main text is about 8.43 mm/kV . This result shows a good agreement with that in the main-text, although the nonlinear susceptibility and the material absorption are slightly different in higher frequencies.

In summary, we can conclude that the measured signal indeed comes from the DFG nonlinear process, rather than from the pump laser directly by the comparison of Experiment 1, Experiment 2, and the main-text experiment. Furthermore, in the theoretical view, as we explained in last Response, the influence of pump laser on the DFG signals can also be excluded from the following aspects:

“First, THz waves generated by pump laser (ISRS) have a broadband spectrum, which is centered at about 0.5 THz, seen in our previous work, [Ref. 41 (Q Wu, et al., Opt. Express 17,

9219 (2009)]. However, in the tilted-wavefront generation, only the frequency components that match the waveguide modes remain, as shown in Fig. 4(b), where even the energy distribution at about 0.5 THz is still very low. If the pump laser could generate THz waves at the difference frequency, it must also generate THz waves at the frequency range between the two waveguide modes, especially at 0.5 THz, rather than only at the difference frequency point.

Second, even if we assume ISRS could generate THz waves at the difference frequency point, it is different from the other two frequencies that match with the waveguide modes, where the generated THz waves could have a continual increase as a function of position x . Considering the general generation efficiency [Ref. 40 (T. Feurer, et al., *Annu. Rev. Mater. Res.* 37, 317 (2007)), and Ref. 41 (Q Wu, et al., *Opt. Express* 17, 9219 (2009))], a 400 μ J pump laser beam could generate THz waves with a maximum amplitude of 0.4 kV/mm focused by a cylindrical lens to a line of about 0.1 mm. Here we consider an extreme condition that the wavefront-tilted pump laser excites THz waves simultaneously in the LiNbO₃ waveguide, since the length of the pump laser on LiNbO₃ is much larger (about 0.9 mm) than that in Ref. 40 (about 0.1 mm), the field amplitude of the pump is about 1/9 of that in Ref. 40. Accordingly, the THz intensity it generated is only $1/9^2$ of that value [Ref. 40 (T. Feurer, et al., *Annu. Rev. Mater. Res.* 37, 317 (2007))], which is about 0.005 kV/mm. Considering our practical setup, where the wavefront-tilted pump does not generate THz waves in a very short time span and for a very large waveguide mode attenuation, the practical THz waves at difference frequency point are far smaller than 0.005 kV/mm, which is negligible compared with the DFG generated value of about 0.935 kV/mm (see Table S1 in the Supplementary Materials Page 2 Line 57, or in the main text Page 5 Line 169).”

According to the above discussions and the experimental fact, the contribution of pump laser generation at the DFG frequency is excluded without ambiguity. In the following response, we will explain the other concerns from the reviewer point by point.

2. I plot the dispersion relation curve of the velocity match of the laser pulses (in black dotted line) on Fig. 2c as follows. The signal, not supported by any waveguide mode, also lies on the line of the velocity match of the laser pulses. Since it lies on the matching line of the pump laser, it is possible that the signal of this mode can be continually contributed from the pump laser pulses. And the effect of the pump laser pulses could last until $x = 2.39$ mm which is within the analysis of the non-waveguide mode signals. Fig. S1 reveals that E_0 increases until $x_3=2.39$ mm, which implies that the contribution from the pump laser could last until $x_3=2.39$ mm. But E_d only shows significant signals at $x_1=1.18$ and $x_2=1.82$ mm. According to the fitting parameters in Eq. (S4) that $\Gamma= 1.28 \text{ mm}^{-1}$ and $d=0.65$ mm, the generation amplitude drops to $\sim 0.221G$ at $x=1.82$ mm compared with the peak value IG at $x=0.65$ mm. It seems that within 2 mm, the effect of pump laser still remains until E_d disappears.

Response2: In this comment, the reviewer analyzed the dispersion relation and the decay of the laser pulses, and concluded that the laser is possible to make contribution to E_d generation.

As we explained in **Response1**, both the experimental and theoretical results can prove that the laser contribution to the DFG signal is ignorable. The black dotted line plotted by the reviewer shows the laser dispersion. The fact that DFG signal lies in this line represents that the DFG signal has the same phase velocity with the laser and the two waveguide modes. This is because in our experiment, the two waveguide modes are excited by modulating the tilt wavefront of the laser, and the DFG signal is generated by the nonlinear frequency-mixing process of the two waveguide modes. Consequently, all of them share the same phase velocity under the requirements of the phase-match condition. This point is also verified by an important experimental fact that the THz waves are *only* generated at the difference frequency point. Therefore, the fact that the DFG signal lies in the laser dispersion curve does *not* indicate the DFG signal comes directly from the laser pulses.

Moreover, the reviewer believes that the laser can generate waveguide modes within $x = 2$ mm by analyzing the laser decay, but here an important factor needs to be carefully considered, the *walk-off effect* between the two waveguide modes. In the nonlinear frequency-mixing process, the generation of the DFG signal is not only depended on the field amplitude of the two waveguide modes, but also requires the two waveguide modes has a good phase match and enough energy overlap (no walk-off effect). The walk-off between the two waveguide modes is explained in the main text, and thus the DFG signal stops being generated after the two waveguide modes walk off.

3. In addition, if $\chi^{(2)}$ at THz range is so giant, why didn't second harmonic generation of v_0 significantly appear in this result? As mentioned by the authors, the DFG process $\chi^{(2)}$ ($v_d = v_1 - v_0$) encounters the walk off effect because of different group velocities. However, the SHG $\chi^{(2)}$ ($2v_0 = v_0 + v_0$) should not suffer from the walk-off effect. The square of E_0 is also larger than the product of E_0 and E_1 . As a result, signals from SHG is expected to be larger than that from DFG if $\chi^{(2)}$ of these two processes are on the same orders. But in Fig. 4b, it is puzzling that the signal at $2v_0 = 0.7$ THz is much weaker than that of $v_d = 0.76$ THz.

Response3: The reviewer comments that the missing of the SHG signal is unreasonable in our experiment by analyzing the intensity of the waveguide mode ν_0 and the walk-off effect. As is well known, the intensity of the nonlinear generated signal depends on the intensity of the incident field, the *phase-match condition*, and the walk-off effect. The missing of the SHG signal is because the phase-match condition is not well satisfied here.

In THz regime, the anisotropic property of the LiNbO₃ crystal/waveguide is also a better choice to realize the SHG at THz frequency. In our previous work, [Ref. 44, Yang, C. et al., Opt. Express 18, 26351 (2010).], the anisotropic effective phase refractive indices of TE waveguide modes were investigated. In LiNbO₃ crystal, the SHG needs to carefully select the excitation angle to satisfy the phase-match condition. In Fig. R4, we show the diagram of SHG phase-match angle in the refractive index ellipsoid of LiNbO₃ crystal, where the black dashed line shows the SHG phase-match angle.

Fig. R4 Diagram of SHG phase match in LiNbO₃ crystal. The red circle and ellipse indicate the refractive indices of ordinary and extraordinary light for THz waves with frequency ν_0 , while the blue circle and ellipse indicate that for SHG signal with frequency $2\nu_0$, respectively. The black dashed line shows the SHG phase-match angle.

In our experiment, we employ the experimental setting proposed by Lin et al. [Ref. 43, Lin, K. H. et al., Appl. Phys. Lett. 95, 103304 (2009).], and fully take advantages of the tilted-wavefront pump-pulses and LiNbO₃ waveguide to satisfy the phase match condition in DFG process along c-axis.

However, because the evident deviation between the SHG phase-match direction and c-axis (Fig. R4), the generation along the z-direction in our experiment cannot satisfy the SHG phase-match condition. Therefore, SHG signal does not appear in our experiment as expected.

4. Appl. Phys. Lett. 99, 071102 (2011) demonstrated high power tunable multicycle THz pulses also in LN. The pump pulse energy (6 mJ and 35 mJ) is order of magnitude higher

than the pulse energy (0.45 mJ) in this manuscript. It should have stronger THz field that could generate larger $\chi^{(2)}$ signals if $\chi^{(2)}$ at THz range is so giant as the authors claimed. Their results at two different pump power are as follows.

[Redacted]

With high pump energy of 35 mJ, it may have SHG component. However, with 6 mJ, the SHG frequency component is not obvious since the power dependence of $\chi^{(2)}$ signal is nonlinear. In contrast, it is very surprising that the authors in this manuscript observed “nonlinear signals” with pump energy as low as 0.45 mJ. It seems that the $\chi^{(2)}$ for SHG at THz range from the results of APL 99, 071102 (2011) is much lower than the $\chi^{(2)}$ for DFG at THz range the authors claimed in this manuscript.

Response4: The reviewer comments that the giant nonlinear susceptibility $\chi^{(2)}$ does not agree with the previously reported results in Appl. Phys. Lett. 99, 071102 (2011). This work by Prof. Nelson’s group mainly focused on the generation of the narrowband THz waves, where the weak SHG signal neither represents the $\chi^{(2)}$ for SHG is smaller than that for DFG, nor indicates the $\chi^{(2)}$ in their system is smaller than that in ours. As we mentioned in **Response3**, phase match condition for SHG needs to be carefully selected in order to obtain a larger SHG signal, but this condition is not taken into account in Appl. Phys. Lett. 99, 071102 (2011). This is also the reason why the SHG signal in that paper seems to be lower than our DFG signals here. In fact, apart from the above discussion, one cannot expect direct comparison from two different experiments at different groups with different experimental conditions.

5. In summary, the effect of pump laser may last until E_d disappears. Second, it is puzzling that DFG overwhelms SHG in the data of this manuscript. Third, the authors observed “nonlinear signal” with pump pulse energy order of magnitude low compared with the results of APL 99, 071102 (2011). I sincerely hope the authors are convinced that pump-power dependence measurement (even with only two different powers) is necessary to strongly support the argument that contribution to E_d is completely from the nonlinear $\chi^{(2)}$ process of DFG in this manuscript.

Response5: We thank the reviewer again for the critical review and constructive comments, which helped us to deeply think about our results. By another two experimental results and the theoretical analyses, we show that the contribution to E_d is completely from the DFG process and the direct pump pulse contribution is ignorable. The intensity of the nonlinear generated signal depends on the intensity of the incident field, the phase-match condition, and

the walk-off effect. The walk-off effect is ignored in the reviewer's first concern, and the phase-match condition is overlooked in the second and third concerns.

To make these issues clear in response to this referee's comments, we have added a new section in Suppl Mater: "2. Additional results/discussion about pump-power dependence", where we made clear why the observed DFG signal comes from the nonlinear process not from enhanced pump pulse itself, and why the DFG signal seems to be stronger than the SHG signal in our experiment. We have also added the citation of APL 99, 071102 (2011).

After all the concerns from the reviewer is properly explained, we hope our revised manuscript can be accepted for publication in Nature Communications.

Reviewer #3:

The authors have addressed all my concerns in the revisions. While the experimental nonlinear susceptibility was estimated based on four fitting parameters and a few assumptions, such as the "piecewise exponential function" for the pump laser and the monochromatic approximation, I believe the precise detail of the fitting procedure has no significant effect on the order of magnitude of the reported nonlinear susceptibility. Therefore, the authors' claim of a giant nonlinear susceptibility many orders of magnitude larger than previously reported values is valid. This is an interesting experimental result that deserves publication in Nature Communications.

Response: We thank the reviewer very much for his/her positive comments and acceptance of our manuscript. The reviewer has provided many helpful suggestions in the last review. In the revised version, we tried our best to perform a better presentation in the manuscript and explained the questions from all the reviewers.

REVIEWER COMMENTS

Reviewer #2 (Remarks to the Author):

The authors have done the experiments with different pump power and the results seemed to support the nonlinear behaviors of difference frequency generation from the two waveguide mode frequencies. Although why second harmonic generation did not significantly appear in the data was not convincingly addressed, I think this work could be accepted after revision. Details suggestions are listed as follows.

1. The comparison of Fig. S2(b) and Fig. 5(a)[$x_1=1.18$ mm] is nice. However, the description of position x to obtain Fig. S2(b) "the waveguide modes do not walk off" is vague. The DFG component can be seen in Fig. S2(a) but is hardly seen in Fig. S(b). Therefore, the exact position x should be given. If the experimental setup is the same and the pump power is the only factor to change, the comparison should be done at the same position x . Since E_{DFG} is proportional to the product of E_0 and E_1 , quantitative comparison might be done through the model Eq. (S1) – (S6). For example, χ_2 was fitted as 1.58 mm/kV for the pump power condition in the main text. Could this $\chi_2=1.58$ mm/kV and the E_1 ($x=1.18$ mm) and E_0 ($x=1.18$) for low pump power condition be used to predict E_d at 1.18mm? Would this value be indeed in the noise level of the spectrum?

2. The authors suggested that SHG should be observable with phase-matching condition and proposed Type 1 phase match condition that the waveguide effective index of doubled frequency is the same with the effective index of the fundamental frequency. Therefore, they proposed the laser dispersion matching should be along a certain direction between TE and TM mode (Ref. 44) that the double frequency should be the waveguide mode to enhance SHG. However, the authors ignored the fact that the difference frequency component demonstrated in this work is not within the waveguide mode, either. Maybe I can rephrase this puzzling question as follow. Along the laser dispersion line, why do components of $(\nu_1-\nu_0)$ and $(\nu_1+\nu_0)$ prefer to appear compared with $(2*\nu_0)$? If this can not be correctly addressed, the authors should be moderate in the supplementary materials to explain the absence of SHG and confess this is an open question. And in the main text, the authors should also be moderate to state different frequency component is selectively enhanced (compared with SHG) and the χ_2 for DFG was calculated based on the assumption that the effect of pump laser is totally ignored as explained in the supplementary material.

Reviewer #3 (Remarks to the Author):

Reviewer #2 raised three interesting points: First, why is there no SFG at $2\nu_0$? Second, it is important to carry out a power dependence study. Third, and most importantly, the E_d signal might be generated by the pump laser, rather than the DFG process as claimed in the paper.

I think that the authors provided a satisfactory explanation for the absence of SFG at $2\nu_0$, which is due to phase mismatch. The authors also carried out two additional measurements at lower and higher power, showing that the E_d signal is indeed proportional to $|E_0*E_1|$ as expected from a DFG process. These new results strengthen the claim of the paper.

For the last point, the authors offered two convincing answers. The pump laser generates THz wave in a broad range of frequency, but most are attenuated except the components at ν_0 and ν_1 . Note that the decay length due to the attenuation, $1/\Gamma_M$, is only 0.05 nm, thus the components with frequency different from ν_0 and ν_1 die out very quickly. The fact that a significantly large E_d field

survives up to 2mm at uniquely $\nu_1 - \nu_0$, and no other frequencies between ν_0 and ν_1 , indicates that this field is generated by a DFG process of E_0 and E_1 . In addition, the DFG χ^2 extracted from the experiment agrees well with theoretical estimates. Therefore, one can be very confident that the E_d signal comes from DFG.

There is a surprising feature in the new figures that I think needs clarification. In theory the DFG peak at $\nu_1 - \nu_0$ and SFG peak at $\nu_1 + \nu_0$ have the same power dependence: they are both proportional to $|E_0 E_1|$ (see Boyd, Nonlinear Optics). Thus, one expects that these two peaks change by the same ratio when the pump power is raised by 43% from Fig. R3 to Fig. R2 in the authors' response. However, the figures show that the DFG peak increases by only a factor of ~ 1.6 while the SFG peak increases by 4 times. I find this quite puzzling.

Replies to the reviewers' comments:

Reviewer #2 (Remarks to the Author):

The authors have done the experiments with different pump power and the results seemed to support the nonlinear behaviors of difference frequency generation from the two waveguide mode frequencies. Although why second harmonic generation did not significantly appear in the data was not convincingly addressed, I think this work could be accepted after revision. Details suggestions are listed as follows.

Response: Thank the reviewer very much. We are glad that a solid evidence to support the nonlinear behavior of the difference frequency generation was constructed with the help of the reviewer's previous comments. The reviewer thinks the weak appearance of the SHG signal was not given a convincing explanation, which we fully agree after reading the detail comment, so we try to address this comment here (Comment 2). Indeed, there're still open questions merit further investigation and we shall consider them in our further work.

1. The comparison of Fig. S2(b) and Fig. 5(a)[$x=1.18$ mm] is nice. However, the description of position x to obtain Fig. S2(b) "the waveguide modes do not walk off" is vague. The DFG component can be seen in Fig. S2(a) but is hardly seen in Fig. S2(b). Therefore, the exact position x should be given. If the experimental setup is the same and the pump power is the only factor to change, the comparison should be done at the same position x . Since E_{DFG} is proportional to the product of E_0 and E_1 , quantitative comparison might be done through the model Eq. (S1) – (S6). For example, χ^2 was fitted as 1.58 mm/kV for the pump power condition in the main text. Could this $\chi^2=1.58$ mm/kV and the E_1 ($x=1.18$ mm) and E_0 ($x=1.18$) for low pump power condition be used to predict E_d at 1.18mm? Would this value be indeed in the noise level of the spectrum?

Response: Here the reviewer has two suggestions: first, he/she advises us to provide the specific value of x in Fig. S2(b); second, he/she also suggests that we should provide a calculation of the DFG signal according to known $\chi^{(2)}$ and the measured E_0 and E_1 in low pump-power case, to further prove the DFG signal is in the noise level of the spectrum. We agree with the reviewer, and thus we added the corresponding calculation in our manuscript.

First, in Fig. S2(b), the specific value of x is selected as 1.14 mm, in agreement with the main-text experiment, the value has been given in the revised Supplementary Materials (Lines 84~85, Page 4).

Second, according to the calculated $\chi^{(2)}=1.58$ mm/kV and the measured E_0 and E_1 , the DFG signal is calculated and evaluated as follows, which has also been added in the Supplementary Materials (Lines 90~119, Page 5):

According to Fig. S2(b), we can calculate the field amplitude of E_0 and E_1 by a Gaussian fitting (the same method as in the main text), and we obtain

$$E_0(x = 1.14) = 0.74 \text{ kV/mm}$$

$$E_1(x = 1.14) = 0.22 \text{ kV/mm}$$

By using the calculated nonlinear susceptibility, one can evaluate the DFG amplitude by

$$E_d = p\chi^{(2)}E_0E_1 \approx 0.1239 \text{ kV/mm}$$

Here we ignored the propagation of E_d for simplicity. Suppose a temporal expansion of 4 ps (similar to the main-text experiment), we can obtain the relative intensity in the frequency domain.

After calculation, we can get the Fourier spectrum of the DFG signal, which is shown in Fig. R1(a). This value is indeed in the noise level of the Fig. S2(b), which is shown below as Fig. R1(b) with superimposed data from Fig. R1(a) for direct comparison (please notice that the two sub figures have different y-axis scales – differs by one order of magnitude).

Fig. R1. The theoretical evaluation of the Fourier spectrum of E_d according to the main-text nonlinear susceptibility. (a) The Fourier spectrum of evaluated E_d . (b) The calculated E_d was superimposed onto the experimental data for comparison.

2. The authors suggested that SHG should be observable with phase-matching condition and proposed Type 1 phase match condition that the waveguide effective index of doubled frequency is the same with the effective index of the fundamental frequency. Therefore, they proposed the laser dispersion matching should be along a certain direction between TE and TM mode (Ref. 44) that the double frequency should be the waveguide mode to enhance SHG. However, the authors ignored the fact that the difference frequency component demonstrated in this work is not within the waveguide mode, either. Maybe I can rephrase this puzzling question as follow. Along the laser dispersion line, why do components of $(\nu_1 - \nu_0)$ and $(\nu_1 + \nu_0)$ prefer to appear compared with $(2 * \nu_0)$? If this cannot be correctly addressed, the authors should be moderate in the supplementary materials to explain the absence of SHG and confess this is an open question. And in the main text, the authors should also be moderate to state different frequency component is selectively enhanced (compared with SHG) and the χ_2 for DFG was calculated based on the assumption that the effect of pump laser is totally ignored as explained in the supplementary material.

Response: We thank the reviewer for the critical comments and helpful suggestions. In the last response, we did ignore the influence of the waveguide modes on the phase-matching condition. After carefully reading and understanding the reviewer's point, we realized that the missing (or weak appearance at high pump power) of the SHG signal in our experiment is a complicated question, which could depend on several factors that include the phase-match

selection, the transient effect, and the anisotropic subwavelength waveguide modes in the experiments. We totally agree that these issues require further studies.

Therefore, we followed the suggestions of the reviewer and stated clearly that this problem is an open question in the revised manuscript (Lines 256~259, Page 8) and the supplementary materials (Lines 171~176, Page 7).

Reviewer #3 (Remarks to the Author):

Reviewer #2 raised three interesting points: First, why is there no SFG at $2\nu_0$? Second, it is important to carry out a power dependence study. Third, and most importantly, the Ed signal might be generated by the pump laser, rather than the DFG process as claimed in the paper.

I think that the authors provided a satisfactory explanation for the absence of SFG at $2\nu_0$, which is due to phase mismatch. The authors also carried out two additional measurements at lower and higher power, showing that the Ed signal is indeed proportional to $|E_0 \cdot E_1|$ as expected from a DFG process. These new results strengthen the claim of the paper.

For the last point, the authors offered two convincing answers. The pump laser generates THz wave in a broad range of frequency, but most are attenuated except the components at ν_0 and ν_1 . Note that the decay length due to the attenuation, $1/\Gamma_M$, is only 0.05 nm, thus the components with frequency different from ν_0 and ν_1 die out very quickly. The fact that a significantly large Ed field survives up to 2mm at uniquely $\nu_1 - \nu_0$, and no other frequencies between ν_0 and ν_1 , indicates that this field is generated by a DFG process of E_0 and E_1 . In addition, the DFG χ_2 extracted from the experiment agrees well with theoretical estimates. Therefore, one can be very confident that the Ed signal comes from DFG.

Response: We thank the reviewer for the insightful and positive comments.

There is a surprising feature in the new figures that I think needs clarification. In theory the DFG peak at $\nu_1 - \nu_0$ and SFG peak at $\nu_1 + \nu_0$ have the same power dependence: they are both proportional to $|E_0 \cdot E_1|$ (see Boyd, Nonlinear Optics). Thus, one expects that these two peaks change by the same ratio when the pump power is raised by 43% from Fig. R3 to Fig. R2 in the authors' response. However, the figures show that the DFG peak increases by only a factor of ~ 1.6 while the SFG peak increases by 4 times. I find this quite puzzling.

Response: Thank the reviewer for the interesting question. Actually, the changing of the DFG signal was supported by the nonlinear frequency mixing process as seen from the pump-power dependent measurement. In our manuscript, the quantitative analyses for the DFG signal is valid, since it has been verified by both the theory and the experiment. Accordingly, the SFG signal should be much weaker since it has to suffer an obvious absorption as compared to the DFG signal.

The puzzling behavior of the SFG in the high-pump power case seems to be caused by a false signal in our experiment. In our pump-probe system, the probe pulse needs a time period about $t = \Delta\tau + nd_{LN}/c$ to pass through the whole sample and get the phase change caused by THz wave, as illustrated in Fig. R2 (the probe-pulse expansion is ignored). The pulse duration of probe laser is about 120 fs, and the refractive index of LN for the probe is 2.31, the thickness of the LN sample is 50 μm , so the time $t = 505$ ps, which also indicates the

temporal resolution of our system is 505 ps. Therefore, a false signal may appear near $\nu = 1/t = 1.98$ THz. Since the frequency of the SFG signal in the high-pump power is 1.73 THz, near the exceptional point of 1.98 THz, so the absolute value of 4 times enhancement is very likely caused by the false signal. In our manuscript, we have avoided such a false signal by just analyzing DFG signal, which is far away from the exceptional frequency point of 1.98 THz. We have added a brief sentence in Supplementary Materials to address this issue (Lines 176~180, Page 7).

Fig. R2. The dynamic process of phase accumulation of the probe pulse. (a) Probe pulse starts to enter the LN sample (where the data collection starts). (b) probe pulse leaves the LN sample (where the data collection ends). This process takes about 505 ps to finish in a 50 μm -thick LN sample.

We thank again both reviewers for their time and critical review, and hope the revised manuscript can be accepted now.

REVIEWERS' COMMENTS

Reviewer #3 (Remarks to the Author):

I recommend publication of the revised manuscript.

Response to the Reviewers:

Reviewer #3:

I recommend publication of the revised manuscript.

Response: We thank the reviewer very much for his/her positive comments and acceptance of our manuscript. The reviewer has provided many helpful suggestions in the previous reviews.